# ON RECONSTRUCTABILITY OF GRAPH NEURAL NETWORKS

## ABSTRACT

Recently, the expressive power of GNNs has been analyzed based on their ability to determine if two given graphs are *isomorphic* using the WL-test. However, previous analyses only establish the expressiveness of GNNs for graph-level tasks from a global perspective. In this paper, we analyze the expressive power of GNNs in terms of *Graph Reconstructability*, which aims to examine whether the topological information of graphs can be recovered from a local (node-level) perspective. We answer this question by analyzing how the output node embeddings extracted from GNNs may maintain important information for reconstructing the input graph structure. Moreover, we generalize GNNs in the form of *Graph Reconstructable Neural Network (GRNN)* and explore *Nearly Orthogonal Random Features (NORF)* to retain graph reconstructability. Experimental results demonstrate that GRNN outperforms representative baselines in reconstructability and efficiency.

## 1 INTRODUCTION

Graph neural networks (GNNs) (Kipf & Welling, 2017), by jointly leveraging node features and graph structure information, are prominent for representation learning of graph-structured data, which can support critical tasks in various domains, including chemistry (Dai et al., 2019), biology (Gasteiger et al., 2021), social media (Fan et al., 2019), and transportation (Zhou et al., 2020). Specifically, GNNs learn node embeddings by recursively aggregating neighborhood embeddings and transforming features from their neighbors in a message-passing framework (Gilmer et al., 2017). As a result, the learned node embedding can be adopted in node-level tasks, including node classification (Kipf & Welling, 2017), link prediction (Zhang & Chen, 2018), and community detection (Lu et al., 2020). Moreover, by aggregating all the node embeddings in a graph, the graph embedding can also be exploited in many graph-level tasks, e.g., molecular design (Ying et al., 2018).

Despite their empirical success, recent studies are concentrating on the theoretical analysis of the expressive power of Graph Neural Networks (GNNs). For instance, Xu et al. (2018a) show that the message-passing GNNs are no more powerful than the 1-WL test (Weisfeiler & Leman, 1968), i.e., distinguishing whether two graphs are isomorphic. Accordingly, they propose Graph Isomorphism Network (GIN) to approximate the 1-WL test. Due to its limited power (equivalence to the 1-WL test), follow-up efforts exploit the higher-dimensional k-WL test (Morris et al., 2019) and auxiliary node features, e.g., positional encoding (Chen et al., 2022) to improve the capability of GNNs. However, the focus of these studies predominantly remains on the ability of GNNs to distinguish between different graph structures, often overlooking their capacity to preserve structural information, such as node degree (Tang et al., 2021) or adjacency nodes (Chanpuriya et al., 2021).

This concern for structural preservation and enhancement is pivotal, especially when considering the role of contextual information embedded in graph topology. This is particularly notable in social networks (Chen et al., 2021), where the homophily principle (Zhu et al., 2020; Liu et al., 2021) suggests that connected nodes tend to share similar features or labels in various applications. This shared contextual information is crucial for node classification, link prediction, and graph pattern analysis with GNNs. However, current literature focuses mainly on designing GNNs to gather contextual data, overlooking its impact on preserving structural information in graphs. This disconnect calls for further research to address both aspects holistically.

In this paper, we investigate the representational ability of various GNNs to encode the *complete graph topology* in the message-passing process. Specifically, we theoretically analyze the expressive power of GNNs in terms of *Graph Reconstructability*, i.e., whether the learned node embeddings from existing GNNs can reconstruct an input graph. Indeed, reconstructability is closely related to graph expressivity because preserving the underlying topological structure of the graph in the learned representations is critical to the accuracy and efficiency of graph mining tasks, such as link

prediction (Zhang & Chen, 2018) and community detection (Chen et al., 2020c). We examine two representative GNNs, GCN (Kipf & Welling, 2017) and GIN (Xu et al., 2018a), in our analysis via two feature initialization schemes widely used in graph mining applications: 1) *identity features* (Murphy et al., 2019) which encode unique identity information, and 2) *contextual features* (Liu et al., 2021) which encode node attributes.[1]

The theoretical results manifest that GIN is more potent in preserving reconstructability than GCN because GIN properly exploits the self-embedding weighting parameter to control the weight of the central node embedding in the output representation. Moreover, identity features are more effective than contextual features in preserving reconstructability. Our analysis proves that both GCN and GIN can achieve reconstructability with identity features. Nonetheless, they incur significant computational and memory costs in the message-passing process, i.e., the costs increase quadratically with the number of nodes in the graph. In addition, while contextual features may positively affect graph reconstructability, they rely heavily on GNNs' ability to capture certain phenomena, e.g., homophily in social networks, to achieve better graph constructability. In other words, the reconstructability of GNNs with contextual features is overshadowed by their strengths in capturing correlations, e.g., homophily, which undermines the reconstructability of GNN to generalize on disassortative graphs where nodes with the same label are distant.

To address these two issues, we first propose *Nearly Orthogonal Random Features (NORF)* to ensure the orthogonality while keeping the identity of each node in input features by uniformly sampling the feature hypersphere for addressing the scalability issue arising ih identity features. Meanwhile, to simultaneously retain graph reconstructability and boost the efficiency, we generalize existing GNN architectures to the *Graph Reconstructable Neural Network (GRNN)* by relaxing certain constraints on the graph structure, e.g., those tailored for node degree and graph homophily, to significantly reduce the embedding dimensionality to $O(log(|V|))$. Extensive experiments on synthetic and real-world graphs show that the empirical results are consistent with our theoretical analysis. Moreover, we also demonstrate that the ability of the proposed GRNN in network mining tasks, including link prediction and community detection, is achieved by preserving graph reconstructability.

The contributions of this paper are summarized as follows.

- We explore a new line of research on the graph reconstructability for analyzing the capability of GNNs in encoding the complete graph structure in node embeddings under various feature initialization schemes.
- Based on the theoretical results, we propose the *Graph Reconstructable Neural Network (GRNN)*, employing *Nearly Orthogonal Random Features (NORF)* for the initialization of features, which not only improves graph reconstructability but also enhances computational efficiency in $O(log(|V|))$.
- Experimental results manifest that retaining graph reconstructability benefits many network mining tasks, including link prediction and community detection .

## 2 RELATED WORKS

Graph Neural Networks (GNNs) (Kipf & Welling, 2017; Hamilton et al., 2017) have emerged as a powerful tool for various graph mining tasks, including recommendation (Chen et al., 2020a), drug discovery (Wieder et al., 2020), computer vision (Wang et al., 2020), and natural language processing (Tu et al., 2019). Specifically, GNNs leverage the graph topology (Chen et al., 2020b) to derive the node embedding by aggregating its own and neighboring node features through message-passing (Battaglia et al., 2018). Although applying GNNs to different applications has achieved great success, growing interests have recently been drawn to their theoretical limitations. For instance, Morris et al. (2019) and Xu et al. (2018a) point out that the expressive power of message-passing GNNs is at most the same as the 1-WL test (Weisfeiler & Leman, 1968). To improve the expressive power of GNNs, a considerable amount of follow-up effort has been devoted. Maron et al. (2019) propose higher-order variants of GNNs by approximating high-order WL tests. However, they are impractical for real-time applications due to an unacceptable computational cost (Wijesinghe & Wang,

---

[1] There is a related but different idea, graph reconstruction conjecture (Cotta et al., 2021), which aim to investigate graph isomorphism through reconstructing the original graph using sampled subgraphs. In contrast, our idea explores the manifold of node embeddings, where the embedding of connected nodes are closer.

2021). Moreover, Liu et al. (2020) and Chen et al. (2020b) leverage inductive biases by analyzing a set of pre-defined topological features, e.g., triangles and cliques. Nevertheless, prior information regarding graph sub-structures needs to be pre-computed.

Another dimension of expressive power is the ability for GNNs to approximate the permutation-invariant functions (Keriven & Peyré, 2019). However, these models often involve tensors that grow in size with the graph size, making them impractical. Chen et al. (2019) demonstrate a connection between the ability to distinguish non-isomorphic graphs and approximate permutation-invariant functions on graphs. Nevertheless, GNNs commonly used in practice are not capable of universal approximation. Loukas (2019) finds that GNNs under certain assumptions are Turing universal but lose power when their depth and width are limited. Note that these findings rely on the assumption that nodes have distinct features and focus on the relationship between the depth and width of GNNs.

In contrast to the above analyses of GNNs in a graph-level perspective, Zhu et al. (2020) and Liu et al. (2021) analyze the ability of GNNs on semi-supervised node classifications (Kipf & Welling, 2017), based on the assumption of homophily, i.e., most connections occur among nodes with similar features. These studies focus on how GNNs gather information from the graph's topology to predict correct node labels. Chanpuriya et al. (2021) study the connection between the learned embeddings and input graphs via skip-gram model (Mikolov et al., 2013) without considering the impact of the node features and GNN architectures. Zhang & Chen (2018) propose a heuristic by extracting the high-order graph structure for link prediction. However, we evaluate the expressive power of GNNs from a different perspective, i.e., analyzing the abilities of different GNNs to preserve the complete topological information in the embedding under different feature initialization and graph structure.

## 3 PRELIMINARY

Following prior theoretical frameworks (Liu et al., 2021; Xu et al., 2018b), we first analyze two prominent GNNs: *GCN* (Kipf & Welling, 2017) and *GIN* (Xu et al., 2018a), which are widely-adopted as the example backbones to analyze the expressive power of GNNs in the node-level (Xu et al., 2018b) and graph-level (Zhang & Chen, 2018) tasks, respectively. Our theoretical framework can be easily extended for other GNNs with different aggregation schemes, such as other normalization (Wu et al., 2019) and attention (Veličković et al., 2018), as long as they follow a message-passing process.

Generally, GNNs use the graph $G = (V, E)$ and associated node features $\mathbf{X}$, which can be either identity features or contextual features (or combined), to learn a representation vector $\mathbf{h}_i$ for each node $v_i$. Note that these two features are functionally different and thus treated separately in this work. Modern GNNs follow a learning paradigm that iteratively updates the representation of a node by aggregating representations of its first or higher-order neighbors. Let $\mathbf{h}_i^{(k)} \in \mathbf{H}^{(k)}$ denote the representation of $v_i$ at the $k$-th layer, while $\mathbf{h}_i^{(0)}$ is initialized by $\mathbf{x}_i$. GCN averages the neighborhood embeddings, which can be formally defined as follows.

$$\text{GCN}^{(k)}(v_i) = \text{MLP}^{(k)}(\mathbf{h}_i^{(k-1)} + \frac{1}{D_i} \sum_{v_j \in N(v_i)} \mathbf{h}_j^{(k-1)}),$$

where $N(v_i)$ and $D_i$ denote the neighborhood set and the degree of $v_i$. MLP consists of a trainable projection matrix and a activation function to transform the embeddings across layers.[2]

On the other hand, GIN sums up the embedding of the neighborhoods nodes without normalization, which can be formally defined as follows.

$$\text{GIN}^{(k)}(v_i) = \text{MLP}^{(k)}((1 + \epsilon^{(k)})\mathbf{h}_i^{(k-1)} + \sum_{v_j \in \mathcal{N}(v_i)} \mathbf{h}_j^{(k-1)})$$

Note that GIN adopts an irrational number $\epsilon^{(k)}$ to distinguish the self-embedding of central node $v_i$ from the neighborhood embeddings. Therefore, it is able to distinguish a unique tuple $(v_i, \mathcal{N}(v_i))$ from $(v_j, \mathcal{N}(v_j))$ even if the two nodes $v_i$ and $v_j$ share the same neighborhood $N(v_i) \equiv N(v_j)$. Thus, GIN can approximate the 1-WL test, which is the upper bound of the expressive power of those message-passing GNNs in graph-level tasks (Morris et al., 2019).

---

[2]Here, we follow the definition in (Xu et al., 2018b) to simplify the GCN. However, our proof is also held for the original GCN (Kipf & Welling, 2017) or other general GNN architecture as shown in Theorem 3.

# 4 THEORETICAL ANALYSIS FRAMEWORK

Here, we first introduce *Graph Reconstructability* and provide a theoretical framework to analyze whether the embeddings learned in a GNN from node features can reconstruct the node adjacency. Following prior theoretical analyses on GNNs (Liu et al., 2021), we consider different schemes of feature initialization, including identity features and contextual features.

**Definition 1** (Graph Reconstructability). *Graph reconstructability is defined to be the ability of a model to predict the input adjacency matrix from the node features.*

**Proposition 1.** *(Proof in Appendix A.4) A model is graph reconstructable if and only if the learned embeddings are able to distinguish linked and unlinked node pairs, where the inner product of the embeddings of a linked node pair is strictly greater than that of an unlinked node pair, i.e.,*

$$\mathbf{h}_i^\top \mathbf{h}_j > \mathbf{h}_i^\top \mathbf{h}_k, \forall (v_i, v_j) \in E \text{ and } (v_i, v_k) \notin E.$$

Based on Proposition 1, we can determine if the node embeddings learned by a GNN can reconstruct a graph using the inner product of embeddings for linked and unlinked node pairs. Note that a logistic classifier trained through gradient descent can be employed to extract this decision boundary.

## 4.1 IDENTITY FEATURES INITIALIZATION

First, we evaluate the reconstructability of GNN models by exploring the identity features (Murphy et al., 2019), i.e., the node index represented in one-hot encoding.

**Definition 2** (Identity Features (IF)).

$$x_i = [x_{i,k}]_{k \in [1,|V|]}, \text{ where } x_{i,k} = \begin{cases} 1 & k = i \\ 0 & otherwise. \end{cases}$$

Only one element is set as $1$ according to the node index, and the rest are set as $0$, which implies that the initial feature of each node is orthogonal to other nodes and thus able to identify itself uniquely. Following Sato et al. (2021), we assume that the graph $G = (V, E)$ is a bounded degree graph with the maximum degree $D \geq 2$. Then, we evaluate the reconstructability of a GNN model by checking whether it can distinguish linked node pairs and unlinked node pairs by Proposition 1.

**Proposition 2.** *With identity features, GCN is provable to distinguish linked and unlinked node pairs.*

**Proposition 3.** *With identity features, GIN is provable to distinguish linked and unlinked node pairs, if $\epsilon > \frac{D}{2} - 1$.*

Proposition 2 demonstrates that GCN retains reconstructability unconditionally by assigning linked node pairs a value of $1$ and normalizing the rest of the neighborhood to $\frac{1}{D}$. Consequently, the inner product between unlinked nodes is less than or equal to $\frac{1}{D}$, while that of linked nodes exceeds $\frac{2}{D}$. In contrast, GIN does not normalize the value through the degree. By controlling $\epsilon > \frac{D}{2} - 1$, GIN preserves the identity information by considering the proportion of self-embedding of the central node in the output. However, a large $\epsilon$ may sacrifice auxiliary neighborhood information, potentially leading to performance degradation in various tasks, such as node classification.

## 4.2 CONTEXTUAL FEATURES INITIALIZATION

The identity features merely include node identity information. By contrast, real-world graphs usually contain rich contextual information in nodes and within the graph topology (Kipf & Welling, 2017), which empowers the model to process the downstream tasks. In the following, we analyze whether the contextual features are able to encode the topological information for graph reconstructability. Accordingly, we introduce the *homophily ratio* (Zhu et al., 2020) of a graph as follows.

**Definition 3** (Homophily Ratio $\rho$). *Given a graph $G = (V, E)$, where each node $v_i \in V$ is associated with a label $c_i$, the homophily ratio is the fraction of edges that connect nodes of the same label, i.e., $\rho = \frac{1}{|E|} \sum_{e_{ij}} \mathbb{1}(c_i \equiv c_j)$. $\mathbb{1}$ is the indicator function.*

The aggregation process of a GNN model generally exploits the graph homophily by smoothing a node's representation with its neighbors, which focuses on collecting meaningful information for

the node classification tasks. In contrast, in this paper, we analyze whether the learned embeddings contain sufficient information to reconstruct the original graph via the contextual features.

**Assumption 1.** *Each node $v_i$ has a probability $\rho$ that its neighborhood nodes are in the same class. Assuming uniform distribution, the probability for a neighbor node in a given different class is $\frac{1-\rho}{|C|-1}$.*

Following Ma et al. (2022), the contextual features include both signal (label information) and noise. Let the feature be initialized in the fashion of one-hot label encoding, where the label $c_i$ of node $v_i$ is one of the $|\mathcal{C}|$ classes. We define the (noisy) contextual features as follows.

**Definition 4** (Contextual Features (CF)).

$$x_i = [x_{i,k}]_{k \in [1,|C|]}, \text{ where } x_{i,k} \sim \begin{cases} \mathcal{N}(1, \sigma_1) & k = c_i \\ \mathcal{N}(0, \sigma_0) & otherwise, \end{cases}$$

*where noises are introduced in the dimension of value one by $\sigma_1$ and all the others by $\sigma_0$.*

With contextual features, we analyze the reconstructability of GCN and GIN.

**Proposition 4.** *With contextual features, GCN is not provable to distinguish linked and unlinked node pairs if $\sigma_0 > 2\sigma_1$.*

**Proposition 5.** *With contextual features, GIN is provable to distinguish linked and unlinked node pairs, if $\rho > \frac{1}{|C|}$ and $\epsilon > max(|C| - 1, 2(1 + \rho)D)$.*

As indicated in the above propositions, the reconstructability of GNNs may be undermined by noises in contextual features and highly correlated to the distribution of neighborhood contextual features. When feature similarity disregards graph topology (low homophily ratio), GNNs are inclined to struggle with node classification due to a lack of correlated contextual information (Liu et al., 2021; Chen et al., 2022). First, GCN cannot distinguish the linked and unlinked node pairs under significant noise because the similarity of node features becomes independent of the node adjacency.[3] By contrast, GIN exploits $\epsilon$ to preserve the central node's identity and distinguish the linked and unlinked node pairs by directly transmitting the contextual information via propagation. As a result, GIN provides more expressiveness than GCN in the graph-level aspect by the WL test (Liu et al., 2020; Xu et al., 2018a) but also in a local view. Proposition 5 shows that the central node has to reside in the majority class of the neighborhood nodes for homophily, i.e., $\rho > \frac{1}{|C|}$, to achieve graph reconstructability. Besides, the contextual features need to ensure $\epsilon$ to be greater than the number of opposite class $|\mathcal{C}| - 1$ to alleviate the noise from the heterophyllous neighborhoods.

## 5 GENERALIZED GNN FOR RECONSTRUCTABILITY

While identity features can preserve topological information due to their orthogonality, scaling up the identity features in a large graph requires a quadratic-growing overhead in memory and processing to maintain the ability to reconstruct the graph. On the other hand, contextual features containing application-dependent information, such as homophily, do not perform well on a disassortative graph. Therefore, we introduce *Graph Reconstructable Neural Network (GRNN)* with *Nearly Orthogonal Random Feature (NORF)* to generalize a family of GNNs.[4]

### 5.1 NEARLY ORTHOGONAL RANDOM FEATURES

Although identity features can preserve the topological information, the time complexity for processing identity features grows quadratically in the order of $O(|V|^2)$. On the other hand, contextual features are constrained by the homophily ratio of graphs (Zhu et al., 2020; Liu et al., 2021) and thus fail to generalize to disassortative graphs. To address the scalability issue of identity features, we first introduce *Nearly Orthogonal Random Features (NORF)*, which require a much smaller dimensionality than identity features, without compromising much orthogonality among features.

**Definition 5** (Nearly Orthogonal Random Features (NORF)).

$$|\mathbf{x}_i^\top \mathbf{x}_j| < \delta, \forall v_i, v_j \in V,$$

*where the inner product of any pair of nodes is smaller than an orthogonality threshold $\delta \to 0$.*

---

[3]Note that, by setting $\sigma_0 = 0$ , GCN can retain graph reconstructability if $\rho > \frac{1}{|C|}$.

[4]The comparison betweeb GRNN and other graph learning frameworks can be found in Appendix C.

According to the high dimensional statistics, there exist $O(exp(d))$ pairs of orthogonal random features with dimensionality $d$ (Ball et al., 1997). With a proper setting of $d(< |V|)$, $O(exp(d))$ can be much greater than $O(|V|)$. Therefore, we can generate the random features by uniformly sampling $|V|$ embedding vectors from a unit ball as our initial features (Vershynin, 2010).

**Theorem 1.** *With nearly orthogonal random features, GCN is provable to distinguish linked and unlinked node pairs if $\frac{1}{8D} > \delta$.*

**Theorem 2.** *With nearly orthogonal random features, GIN is provable to distinguish linked and unlinked node pairs, if $\epsilon = \frac{D}{2}$ and $\frac{1}{4D^2} > \delta$.*

Equipped with NORF, GNNs are provable to preserve complete graph topology through the aggregation process. Compared to identity features with perfect orthogonality between nodes, NORF can effectively maintain the graph reconstructability by adjusting the degree of $\delta$. The dimensionality of NORF ($O(log(|V|))$) is notably smaller than that of identity features ($O(V)$).

## 5.2 GRAPH RECONSTRUCTABLE NETWORK

After carefully examining the conditions for the maximal reconstructability of the initial features for representative GNNs in Section 4, we propose *Graph Reconstructable Network (GRNN)* by generalizing existing GNN architectures. We adopt the self-embedding weight, motivated by an observation that selecting self-embedding weight $\epsilon$ depends on the maximum degree $D$. Nevertheless, a clear trade-off exists between the orthogonality $\delta$ and maximum degree, denoted as $\frac{1}{4D^2}$. Generally speaking, a larger $\epsilon$ set to preserve graph reconstructability necessitates a higher embedding dimensionality to assure the orthogonality of random features. Nevertheless, GRNN, which is more computationally efficient for larger graphs, is able to relieve the requirement of $\epsilon$.

**Theorem 3.** *With nearly orthogonal random features, GRNN in the following form,*

$$\text{GRNN}^{(k)}(v_i) = \text{MLP}((1 + \epsilon)\mathbf{h}_i^{(k-1)} + \sum_{v_j \in N(v_i)} w_j \mathbf{h}_j^{(k-1)}),$$

*is provable to distinguish linked and unlinked node pairs, if $\epsilon = \frac{\|\mathbf{w}\|_1}{2}$ and $\frac{4}{13\|\mathbf{w}\|_1^2} > \delta$.*

It is worth noting that popular aggregation functions (Kipf & Welling, 2017; Xu et al., 2018a) are special cases of our GRNN by setting different $w_i$, which can be predefined or learned. For example, GCN aggregates the normalized neighborhood features, where the aggregation weight $\|\mathbf{w}\|_1$ is bounded by 1. The weight of GIN is bounded by the maximum degree $D$. Our framework can also support the attention-based GNN. For instance, GAT (Veličković et al., 2018) estimates the aggregation weight $w_j$ by a sharable attention vector $\mathbf{a}^\top[\mathbf{h}_i, \mathbf{h}_j]$, while Graph Transformer Network (Kreuzer et al., 2021) calculates the weight via the bilinear attention $\mathbf{h}_i^\top \mathbf{W}_{Q,K} \mathbf{h}_j$, where their weights are normalized to $O(1)$ by softmax function. As shown in Theorem 3, with different weight matrices, aggregation schemes can provably retain graph reconstructability by controlling $\epsilon$ to balance the orthogonality of initial random features according to the aggregation weight. Other feature initialization, such as identity and contextual features, can also be used in GRNN.

Last, we conclude the requirement of embedding dimensionality to ensure the orthogonality of NORF.

**Corollary 1.** *By uniform sampling $|V|$ embedding vectors as the nearly orthogonal random features, the dimensionality has to be set in the order of $O(\|\mathbf{w}\|_1^4 log(|V|))$ to retain graph reconstructability.*

GRNN notably reduces the embedding dimensionality, leading to better computational efficiency. By Corollary 1, the embedding dimensionality of GRNN is set in the order of $log(|V|)$, which is much smaller than that of identity features, i.e., $O(|V|)$. While the dimensionality grows quartically by the aggregation weight $\|\mathbf{w}\|_1$, GIN requires a larger dimensionality to retain the reconstructability because its aggregation weight is bounded by the maximum degree $D$. Therefore, it is promising to employ the normalized weight to find a lower dimensionality in the order of $O(log(|V|))$. The time complexity of GRNN can be reduced to $O(|E|log(|V|) + |V|log^2(|V|))$ with NORF for retaining graph reconstructability efficiently (detailed in Appendix A.15).

## 5.3 CONNECTION WITH GRAPH MINING TASKS

In the context of GNNs, there is a common practice of fine-tuning embeddings for link prediction, as exemplified in prior studies (Zhang & Chen, 2018). This optimization goal aims to amplify subtle

topological differences between two similar nodes without an edge while reducing disparities between nodes connected by an edge (Srinivasan & Ribeiro, 2019). In contrast to typical link prediction tasks that assume incomplete graphs with missing connections, GRNN operates under the assumption that it can successfully reconstruct connected and unconnected node pairs. If the model successfully reconstructs and distinguishes all existing linked and unlinked node pairs, the scores assigned to unlinked node pairs that should have been linked (removed positive links) would be lower than those of previously linked node pairs (unremoved positive links). Still, in Proposition 6, we can follow a similar proving paradigm where the embeddings of closely connected nodes are typically deemed similar, resulting in a higher value of their inner product, serving the link prediction objective.

**Proposition 6.** *For link prediction, the level of graph reconstructability can be evaluated by the AUC score of the inner product of learned embeddings from GRNN.*

More specifically, if $AUC = 1$, the graph is perfectly reconstructable i.e., none of the linked node pairs has a smaller value of inner product than unlinked ones. Previous studies also employ several pre-training techniques (Hu et al., 2020a;b) to learn node embeddings in a semi-supervised manner. In a node-level pre-training, the objective is to preserve the structural information of the graph in the learned node embedding by measuring the similarity between a pair of nodes (Lu et al., 2021). They typically utilize a skip-gram objective with negative sampling (Mikolov et al., 2013) to ensure the proximity of the nodes. Thus, it leads to the minimization of topological variations within closely interconnected communities in the node representation.

This procedure can be efficiently obtained through the message-passing process in GRNN, because for node classification in assortative graph and community detection, the nodes with more common neighborhoods are more likely to derive similar features from their neighborhoods, leading to closed embeddings. It can also be expressed as the estimation of the affiliation matrix for community detection. We summarize the result in Proposition 7.

**Proposition 7.** *For community detection, the learned embeddings from GRNN approximate the affliction matrix* $\mathbf{H}$ *via symmetric nonnegative matrix factorization.*

# 6 EXPERIMENT

We evaluate graph reconstructability of different GNNs on both synthetic and real graphs. We further conduct link prediction and community detection to demonstrate the superiority of GRNN. Additional experimental results, including running time and node classification, are presented in Appendix B.

## 6.1 SYNTHETIC GRAPHS

Since it is difficult to investigate the impact of important factors on graph reconstructability in a controlled manner with real graphs, we first present experimental results on synthetic graphs. Without loss of generality, we consider the CSBM (Deshpande et al., 2018) model with two classes $c_0$ and $c_1$, where the nodes in the generated graphs are divided into two disjoint sets, $c_0$ and $c_1$, corresponding to the two classes, respectively. Edges are generated for two nodes in the same classes with an intra-class probability $p$ and for two nodes in different classes with an inter-class probability $q$. Therefore, the homophily ratio $\rho$ can be expressed by $\frac{p}{p+q}$. We generate $100,000$ graphs with the sizes ranging from $100$ to $100,000$. As a performance measure, we use *Graph Reconstructable Ratio (GRR)*, which indicates the percentage of graph instances reconstructable from the output embeddings, i.e., $\frac{\#\text{reconstructable graphs}}{\#\text{total graphs}}$.

Against `GCN` and `GIN`, we evaluate three variants of GRNN with different aggregation schemes, including , $\text{GRNN}_{\text{mean}}$, $\text{GRNN}_{\text{max}}$, and $\text{GRNN}_{\text{attn}}$. Furthermore, we adopt the skip-gram objective with negative sampling (Hu et al., 2020a) to train the projection weight without the ground-truth labels.

**Homophily Ratio** $\rho$**.** Figure 1(a) evaluates GRNN with different feature initialization by varying homophily ratio $\rho$ incrementally. When the homophily ratio is small (e.g., $0.2$), contextual features (CF) exhibit low GRR scores due to their strong correlation with the homophily ratio, adversely affecting graph reconstructability. Specifically, when $\rho < 0.5$, i.e., the central node is not in the majority class of neighborhood nodes, GRR scores drop below $0.2$, indicating suboptimal encoding of graph topology in disassortative graphs. In contrast, a higher $\rho$ significantly enhances GRR scores, aligning with Propositions 4 and 5. Notably, the homophily ratio does not impact the results of identity features (IF) and NORF, as they encode the graph in a (nearly) orthogonal manner.

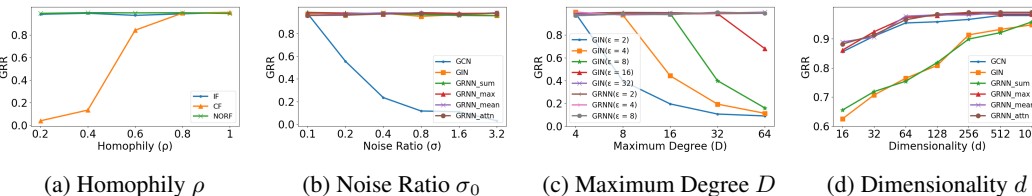

|  | (a) Homophily $\rho$ | (b) Noise Ratio $\sigma_0$ | (c) Maximum Degree $D$ | (d) Dimensionality $d$ |

Figure 1: Quantitative results on the synthetic graphs (GRR).

Table 1: Graph reconstructability on the real-world graphs (AUC).

| method | Pubmed ($\rho = 0.80$) | | | | Actor ($\rho = 0.22$) | | | |
| --- | --- | --- | --- | --- | --- | --- | --- | --- |
| | IF | CF | NORF | NORF + CF | IF | CF | NORF | NORF + CF |
| GCN | 0.9981 | 0.9877 | 0.9632 | 0.9892 | 0.9963 | 0.7491 | 0.9670 | 0.9721 |
| GIN($\epsilon = 1$) | 0.8987 | 0.8889 | 0.8177 | 0.9102 | 0.7934 | 0.7004 | 0.7938 | 0.8221 |
| GIN($\epsilon = \frac{D}{2}$) | 0.9983 | 0.9894 | 0.9183 | 0.9959 | 0.9984 | 0.7510 | 0.9021 | 0.9112 |
| GAT | 0.9919 | 0.9766 | 0.9641 | 0.9902 | 0.9723 | 0.7001 | 0.9516 | 0.9701 |
| SGC | 0.9812 | 0.9782 | 0.9429 | 0.9821 | 0.9919 | 0.7212 | 0.9560 | 0.9624 |
| GRNN$_{\text{mean}}$ | 0.9981 | 0.9832 | 0.9854 | 0.9912 | 0.9943 | 0.7590 | 0.9863 | 0.9924 |
| GRNN$_{\text{max}}$ | 0.9902 | 0.9835 | 0.9561 | 0.9913 | 0.9865 | 0.7621 | 0.9799 | 0.9829 |
| GRNN$_{\text{attn}}$ | 0.9986 | 0.9847 | 0.9841 | 0.9903 | 0.9966 | 0.7423 | 0.9835 | 0.9912 |

**Noise Ratio $\sigma_0$.** Next, we evaluate the impact of noise on contextual features. We introduce different levels of noise to the contextual features on the zero value dimension by varying $\sigma_0$ while fixing $\sigma_1 = 0.1$. Figure 1(b) shows that all GRNN models and GIN effectively retain the graph reconstructability with different level of noise, while GCN plunges when the level of noise increases, i.e., $\sigma_1 > 2\sigma_0$, consistent with Proposition 4. By carefully controlling $\epsilon$, GIN can reconstruct the original input graph under significant noise according to Proposition 5, but it's still affected by node degree, which is agnostic to GRNN.

**Maximum Degree $D$.** Figure 1(c) examines how the maximum degree $D$ of the input graph affects GRNN and GIN, both using the self-embedding weight $\epsilon$. Theorem 2 underscores the importance of setting $\epsilon$ based on $D$ to maintain GIN's graph reconstructability. GIN's GRR score sharply declines when $\epsilon$ is smaller than $\frac{D}{2}$, making it challenging to preserve graph reconstructability. However, increasing $\epsilon$ necessitates reducing $\delta$ for NORF orthogonality, which, in turn, requires a larger embedding dimension for GIN. In contrast, GRNN remains degree-agnostic and can handle large graphs due to its thoughtful aggregation weight design. Note that choosing an excessively large $\epsilon$ can harm node classification performance, as it may lead to the loss of vital neighborhood information.

**Embedding Dimensionality $d$.** Figure 1(d) analyzes how dimensionality ($d$) in NORF affects the examined models. The findings show that increasing dimensionality enhances orthogonality (Edmonds, 1965), improving graph reconstructability, as shown in Corollary 1. Notably, GIN require larger dimensionality due to the unnormalized aggregation weight $\mathbf{w}$. However, when the aggregation weight is small, GNNs struggle to gather neighborhood information. Setting the aggregation weight to zero reduces message-passing GNNs to MLPs, resulting in decreased performance, especially on assortative graphs.

## 6.2 REAL-WORLD GRAPHS

Here, we conduct experiments on real-world graphs. Following Liu et al. (2021), we evaluate three GRNN variants against GCN, two GIN variants, GAT (Veličković et al., 2018) and SGC (Wu et al., 2019), coupled with various feature initialization schemes, on both assortative (**Pubmed**) (Kipf & Welling, 2017) and disassortative (**Actor**) (Rozemberczki et al., 2021) graphs. We adopt the real node attributes as contextual features. $\epsilon$ is set according to the statistics of each dataset based on our theoretical results, i.e., $\frac{D}{2}$ for GIN and $\frac{\|\mathbf{w}\|_1}{2}$ for GRNN. The embedding dimensionality is set to 256.

It is not feasible to distinguish the model capability in the real-world graph by GRR because each dataset contains one instance and thus only returns 1 or 0. Therefore, we adopt the AUC score as the metric to evaluate the model performance. Specifically, we train an additional logistic classifier to predict the probability of each edge between nodes and rank the results to derive the AUC score. When AUC is 1, the model can correctly distinguish linked and unlinked pairs, i.e., perfectly reconstructable.

Table 2: Link Prediction (AUC).

| method | IF | CF | NORF | NORF +CF |
|---|---|---|---|---|
| GCN | 0.9188 | 0.9192 | 0.9071 | 0.9206 |
| GIN | 0.9284 | 0.9133 | 0.9164 | 0.9291 |
| GAT | 0.9192 | 0.9184 | 0.9174 | 0.9273 |
| SGC | 0.9217 | 0.9219 | 0.9188 | 0.9215 |
| SEAL | 0.9218 | 0.9217 | 0.9198 | 0.9292 |
| GRNN$_{mean}$ | 0.9343 | 0.9244 | 0.9151 | 0.9471 |
| GRNN$_{max}$ | 0.9365 | 0.9290 | 0.9252 | 0.9438 |
| GRNN$_{attn}$ | 0.9388 | 0.9231 | 0.9248 | 0.9511 |

Table 3: Community Detection (Acc.).

| method | IF | CF | NORF | NORF + CF |
|---|---|---|---|---|
| GCN | 0.9210 | 0.9430 | 0.9219 | 0.9340 |
| GIN | 0.9229 | 0.9312 | 0.9102 | 0.9419 |
| GAT | 0.9198 | 0.9300 | 0.9155 | 0.9488 |
| SGC | 0.9213 | 0.9219 | 0.9142 | 0.9511 |
| CommDGI | 0.9323 | 0.9332 | 0.9101 | 0.9601 |
| GRNN$_{mean}$ | 0.9512 | 0.9522 | 0.9290 | 0.9621 |
| GRNN$_{max}$ | 0.9652 | 0.9529 | 0.9321 | 0.9671 |
| GRNN$_{attn}$ | 0.9566 | 0.9425 | 0.9238 | 0.9635 |

Table 1 manifests that identity features maintain the graph reconstructability well for both assortative and disassortative graphs with a high AUC score. Meanwhile, as contextual features tend to collect the distribution of neighborhood features which dominate the topological information, it fails to preserve the reconstructability on a disassortative graph. The condition in Proposition 4 and 5, i.e., $\rho > \frac{1}{|C|}$, does not hold on disassortative graphs and thus undermines the reconstructability.

Finally, we discuss the use of NORF in examined models. GIN with NORF have a weak performance because they require a higher dimensionality to ensure the orthogonality (see Corollary 1). In contrast, GRNN with a normalized weight can retain the graph reconstructability with a smaller dimensionality, i.e., $O(log(|V|))$, where the aggregation weights of GRNN$_{mean}$, GRNN$_{max}$, and GRNN$_{attn}$ are all bounded by 1. In summary, GRNN with NORF achieves competitive performance in all datasets by carefully controlling the aggregation weight $\mathbf{w}$ from neighborhoods and the self-embedding weight $\epsilon$. By combining NORF and contextual features (which would be naturally used in applications), our GRNN achieved the best performance among baselines.

### 6.3 Additional Graph Mining Tasks

**Link Prediction.** To evaluate the applicability of our theoretical framework, we first assess the GRNN model's performance across various feature initialization schemes within the context of link prediction. Following Zhang & Chen (2018), we adopt **Pudmed** dataset and randomly remove $10\%$ existing links as positive testing data. Table 2 presents the model performance for link prediction regarding the AUC score, indicating that contextual features perform similarly to identical features and NORF. This outcome can be attributed to the assortative nature of the PubMed dataset, which enables the preservation of graph reconstructability within the contextual features. Notably, combining NORF and contextual features achieves the highest performance by simultaneously encoding graph structures and contextual features. This enhancement surpasses the performance of previous methods that solely rely on contextual features. Moreover, compared with a state-of-the-art GNN model for link prediction, SEAL (Zhang & Chen, 2018), our GRNN demonstrates superior performance.

**Community Detection.** Following Chen et al., we select the $5,000$ top-quality communities in **DBLP** dataset and then examine nodes in every pair of communities $(\mathcal{C}_i, \mathcal{C}_j)$, with at least one edge across them. We aim to classify whether these nodes belong to $\mathcal{C}_i$ or $\mathcal{C}_j$. We divide the dataset into training and testing sets with a split ratio of $0.8$ and enforce that no community in the testing set belongs to the training set. Table 3 compares the performance of three variations of the GRNN model with GIN and GCN. The results align with our theoretical findings, which suggest that GNNs excel in community detection due to their ability to reconstruct the complete input graph using identity features and NORF. Conversely, the model's performance is hindered by contextual features, when the homophily of the neighborhood distribution has a negative impact on them. Our GRNN also outperforms the community-aware GNN model CommDGI (Zhang et al., 2020) since our GRNN can theoretically retain the complete graph topology.

## 7 Conclusion

In this paper, we introduce a theoretical framework to evaluate the *Graph Reconstructability* of GNNs by analyzing how GNNs encode the graph structure in the node embeddings. We first analyze GCN and GIN with identity and contextual features. Then, we propose *Graph Reconstructable Neural Network (GRNN)* with *Nearly Orthogonal Random Feature (NORF)* to retain the graph reconstructability more efficiently. Future works include measuring the reconstructability of the heterogeneous/knowledge graph models with rich semantics in edges.

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
