# A PROOFS

## A.1 NOTATIONS

Table 4: Table of Notations

| symbol | description | symbol | description |
|--------|-------------|--------|-------------|
| $G(V,E)$ | The input graph | $C$ | The label set of $G$ |
| $D$ | The maximum degree of graph $G$ | $\rho$ | The homophily ratio |
| $\tau_l$ | The probability of linked nodes with a common neighborhood | $\tau_u$ | The probability of unlinked nodes with a common neighborhood |
| $\mathbf{x}_i$ | The initial features of node $v_i$ | $\mathbf{h}_i$ | The processed features after GNN of node $v_i$ |
| $\sigma_0$ | The noise in the dimension of value zero | $\sigma_1$ | The noise in the dimension of value one |
| $\epsilon$ | The self-embedding weight | $\mathbf{w}$ | The aggregation weight |
| $d$ | The embedding dimensionality | $\delta$ | The upper bound of any two features, representing the orthogonality |

## A.2 SUMMARY OF THEORETICAL RESULTS

Table 5: Summary of our theoretical results. $\rho$ is the homophily ratio. $D$ is the maximum degree. $\tau_u$ is the probability that an unlinked pair has a common neighborhood. $\epsilon$ denotes the self-embedding weight. $\delta$ represents the orthogonality of the features.

| | IF ($O(|V|)$) | CF ($O(|C|)$) | NORF ($O(log(|V|))$) |
|--|--------------|---------------|----------------------|
| GCN | any condition | No | $\frac{1}{8D} > \delta$ |
| GIN | $\epsilon > \frac{D}{2} - 1$ | $\rho > \frac{1}{|C|}, \epsilon > max(|C|-1, 2(1+\rho)D)$ | $\epsilon = \frac{D}{2}, \frac{1}{4D^2} > \delta$ |
| GRNN | $\epsilon > \frac{|\mathbf{w}|_1}{2} - 1$ | $\rho > \frac{1}{|C|}, \epsilon > max(|C|-1, 2(1+\rho)\mathbf{w}_1)$ | $\epsilon > \frac{|\mathbf{w}|_1}{2}, \frac{4}{13|\mathbf{w}|_1^2} > \delta$ |

Table 5 summarizes our theoretical results, manifesting that GRNN can preserve graph reconstructability for any feature initialization by carefully adjusting self-embedding weight $\epsilon$ and aggregation weight $\mathbf{w}$. Additionally, with NORF, GRNN minimizes embedding dimensionality $d$ while keeping initial feature orthogonality $\delta$. Note that GRNN can retain the best reconstructability by properly setting the self-embedding weight $\epsilon$ and the aggregation weight $\mathbf{w}$ with the lowest dimensionality in $O(log(|V|))$.

## A.3 DETAILED DISCUSSIONS

From a local (node-level) perspective, the $k$-th layer of a GNN is defined as follows.

$$\mathbf{a}_i^{(k)} = \texttt{AGGREGATE}^{(k)}\left(\left\{\mathbf{h}_j^{(k-1)}\Big| v_j \in N(v_i)\right\}\right), \text{ and } \mathbf{h}_i^{(k)} = \texttt{COMBINE}^{(k)}\left(\mathbf{h}_i^{(k-1)}, \mathbf{a}_i^{(k)}\right),$$

where $N(v_i)$ denotes the neighborhood set of $v_i$. The `AGGREGATE` function is used to gather information from neighbors. Various approaches for `AGGREGATE` have been proposed, such as uniform weight (Kipf & Welling, 2017; Xu et al., 2018a) and neural network (Hamilton et al., 2017; Veličković et al., 2018). On the other hand, the `COMBINE` function fuses the information from neighbors into the self-embedding of the center node.

Following Sato et al. Sato et al. (2021), in the theoretical analysis, the multi-layer perceptron is universal, and the number of dimensions is fixed but allowed to be massive.

**Definition 6.** *The* `MLP` *of GCN and GIN is defined as follows.*

$$\texttt{MLP}(\mathbf{h}) = \texttt{ReLU}(\mathbf{W}\mathbf{h}),$$

*where the projection matrix* $\mathbf{W}$ *is a unitary matrix.*

Following Definition 6 Xu et al. (2018b), we have

$$E[\texttt{MLP}(\mathbf{h}_i)^\top \texttt{MLP}(\mathbf{h}_j)] = \frac{1}{2}E[\mathbf{h}_i^\top \mathbf{h}_j].$$

Thus, it only needs to calculate the inner product of the embedding vectors before activation to evaluate the graph reconstructability.

Following Liu et al. (2021), when two nodes are linked, they have a probability $\tau_l$ with a common neighborhood. Otherwise, the unlinked pair has a probability $\tau_u$ with a common neighborhood. The

above probabilities can be derived according to Robins et al. Robins et al. (2009). Moreover, our theoretical results can be generalized into other models of link, e.g., random-graph, when we are able to derive the probability of link by the stochastic process, e.g., following Wu et al. (2010).[5]

### A.4 PROOF OF PROPOSITION 1

**Proposition 1.** *A model is graph reconstructable if and only if the learned embeddings are able to distinguish linked and unlinked node pairs, where the inner product of the embeddings of a linked node pair is strictly greater than that of an unlinked node pair, i.e.,*

$$\mathbf{h}_i^\top \mathbf{h}_j > \mathbf{h}_i^\top \mathbf{h}_k, \forall (v_i, v_j) \in E \text{ and } (v_i, v_k) \notin E.$$

*Proof.* ($\Rightarrow$) We prove this by contradiction. If the inner product of embeddings for an unlinked node pair is greater than that for any linked node pair, the reconstructed graph has at least one edge different from the input graph. ($\Leftarrow$) One can perfectly reconstruct the input graph with the decision boundary of the linked node pairs and unlinked node pairs. The proposition follows. $\qquad\square$

### A.5 PROOF OF PROPOSITION 2

**Proposition 2.** *With identity features, GCN is provable to distinguish linked and unlinked node pairs.*

*Proof.* After one GCN layer,

$$\mathbf{h}_i = [\mathbf{h}_{i,k}]_{k \in [1,|V|]}, \text{ where } \mathbf{h}_{i,k} = \begin{cases} 1 & k = i \\ \frac{1}{D_i} & v_k \in N(v_i) \\ 0 & otherwise. \end{cases}$$

If two nodes are linked, we have

$$E[\mathbf{h}_i^\top \mathbf{h}_j] = \frac{1}{D_i} + \frac{1}{D_j} + \tau_l \frac{D_i - 1}{D_i D_j}.$$

Otherwise,

$$E[\mathbf{h}_i^\top \mathbf{h}_j] = \tau_u \frac{1}{D_i}.$$

Since

$$\frac{1}{D_i} + \frac{1}{D_j} + \tau_l \frac{D_i - 1}{D_i D_j} > \tau_u \frac{1}{D_i},$$

the theorem follows. $\qquad\square$

### A.6 PROOF OF PROPOSITION 3

**Proposition 3.** *With identity features, GIN is provable to distinguish linked and unlinked node pairs, if $\epsilon > \frac{D}{2} - 1$.*

*Proof.* After one GIN layer,

$$\mathbf{h}_i = [\mathbf{h}_{i,k}]_{k \in [1,|V|]}, \text{ where } \mathbf{h}_{i,k} = \begin{cases} 1 + \epsilon & k = i \\ 1 & v_k \in N(v_i) \\ 0 & otherwise. \end{cases}$$

If two nodes are linked,

$$E[\mathbf{h}_i^\top \mathbf{h}_j] = 2(1 + \epsilon) + \tau_l (D_i - 1).$$

Otherwise,

$$E[\mathbf{h}_i^\top \mathbf{h}_j] = \tau_u D_i.$$

---

[5]The assumption of the probabilities of linked nodes and unlinked nodes that have a common neighborhood is just a simple math trick to reduce the complexity of the equation. Actually, we can assign each node with different values of probabilities of common neighborhood and our theoretical results are also held.

Then,

$$2(1+\epsilon) + \tau_l(D_i - 1) > \tau_u D_i$$
$$\Rightarrow 2(1+\epsilon) + \tau_l D_i > \tau_u D_i$$
$$\Rightarrow 2(1+\epsilon) > (\tau_u - \tau_l)D_i.$$

By setting $\epsilon > \frac{D}{2} - 1$, the above inequality holds. Thus, the theorem follows. $\qquad\square$

## A.7 PROOF OF PROPOSITION 4

**Proposition 4.** *With contextual features, GCN is not provable to distinguish linked and unlinked node pairs if $\sigma_0 > 2\sigma_1$.*

*Proof.* After one GCN layer,

$$h_i = [h_{i,k}]_{k \in [1,|C|]}, \text{ where } h_{i,k} = \begin{cases} \mathcal{N}(1+\rho, (1+\rho)\sigma_1) & k = l \\ \mathcal{N}(\frac{1-\rho}{(|C|-1)}, \sigma_0 + \frac{1-\rho}{(|C|-1)}\sigma_1) & otherwise. \end{cases}$$

If two nodes are linked, we have

$$E[h_i^\top h_j] = \rho((1+\rho)^2 + \frac{(1-\rho)^2}{|C|-1}) + (1-\rho)|C|(\frac{(1+\rho)(\sigma_0 + \frac{1-\rho}{(|C|-1)}\sigma_1)^2 + (\frac{1-\rho}{(|C|-1)})((1+\rho)\sigma_1)^2}{(\sigma_0 + \frac{1-\rho}{(|C|-1)}\sigma_1)^2 + ((1+\rho)\sigma_1)^2}).$$

Otherwise,

$$E[h_i^\top h_j] = \frac{1}{|C|}((1+\rho)^2 + \frac{(1-\rho)^2}{|C|-1}) + (1-\frac{1}{|C|})|C|(\frac{(1+\rho)(\sigma_0 + \frac{1-\rho}{(|C|-1)}\sigma_1)^2 + (\frac{1-\rho}{(|C|-1)})((1+\rho)\sigma_1)^2}{(\sigma_0 + \frac{1-\rho}{(|C|-1)}\sigma_1)^2 + ((1+\rho)\sigma_1)^2}).$$

Suppose that $\sigma_0 > 2\sigma_1$,

$$[\frac{1}{|C|}((1+\rho)^2 + \frac{(1-\rho)^2}{|C|-1}) + (1-\frac{1}{|C|})|C|(\frac{(1+\rho)(\sigma_0 + \frac{1-\rho}{(|C|-1)}\sigma_1)^2 + (\frac{1-\rho}{(|C|-1)})((1+\rho)\sigma_1)^2}{(\sigma_0 + \frac{1-\rho}{(|C|-1)}\sigma_1)^2 + ((1+\rho)\sigma_1)^2})]$$

$$- [\rho((1+\rho)^2 + \frac{(1-\rho)^2}{|C|-1}) + (1-\rho)|C|(\frac{(1+\rho)(\sigma_0 + \frac{1-\rho}{(|C|-1)}\sigma_1)^2 + (\frac{1-\rho}{(|C|-1)})((1+\rho)\sigma_1)^2}{(\sigma_0 + \frac{1-\rho}{(|C|-1)}\sigma_1)^2 + ((1+\rho)\sigma_1)^2})]$$

$$> [\frac{1}{|C|}((1+\rho)^2 + \frac{(1-\rho)^2}{|C|-1}) + \frac{2}{3}(1-\frac{1}{|C|})|C|(1+\rho))] - [\rho((1+\rho)^2 + \frac{(1-\rho)^2}{|C|-1}) + \frac{2}{3}(1-\rho)|C|(1+\rho)]$$

$$= [(\frac{1}{|C|} - \rho)((1+\rho)^2 + \frac{(1-\rho)^2}{|C|-1}) + \frac{2}{3}(\rho - \frac{1}{|C|})|C|(1+\rho))]$$

$$= (\rho - \frac{1}{|C|})(\frac{2}{3}|C|(1+\rho)) - (1+\rho)^2 - \frac{(1-\rho)^2}{|C|-1})$$

$$> 0.$$

Therefore, Proposition 1 doesn't hold. The theorem follows. $\qquad\square$

## A.8 PROOF OF PROPOSITION 5

**Proposition 5.** *With contextual features, GIN is provable to distinguish linked and unlinked node pairs, if $\rho > \frac{1}{|C|}$ and $\epsilon > max(|C| - 1, 2(1+\rho)D)$.*

*Proof.* After one GIN layer,

$$h_i = [h_{i,k}]_{k \in [1,|C|]}, \text{ where } h_{i,k} = \begin{cases} \mathcal{N}(1+\rho+\epsilon, (1+\rho+\epsilon)\sigma_1) & k = l \\ \mathcal{N}(\frac{1-\rho}{(|C|-1)}D_i, \sigma_0 + \frac{1-\rho}{(|C|-1)}\sigma_1 D_i) & otherwise. \end{cases}$$

If two nodes are linked, we have

$$E[h_i^\top h_j] = \rho((1+\rho+\epsilon)^2 + \frac{(1-\rho)^2}{|C|-1}D_i^2)$$

$$+ (1-\rho)|C|(\frac{(1+\rho+\epsilon)(\sigma_0 + \frac{1-\rho}{(|C|-1)}\sigma_1 D_i)^2 + (\frac{1-\rho}{(|C|-1)})((1+\rho+\epsilon)\sigma_1)^2}{(\sigma_0 + \frac{1-\rho}{(|C|-1)}\sigma_1 D_i)^2 + ((1+\rho+\epsilon)\sigma_1)^2}).$$

Otherwise,

$$E[h_i^\top h_j] = \frac{1}{|C|}((1+\rho+\epsilon)^2 + \frac{(1-\rho)^2}{|C|-1}D_i^2)$$
$$+ (1-\frac{1}{|C|})|C|(\frac{(1+\rho+\epsilon)(\sigma_0 + \frac{1-\rho}{(|C|-1)}\sigma_1 D_i)^2 + (\frac{1-\rho}{(|C|-1)})((1+\rho+\epsilon)\sigma_1)^2}{(\sigma_0 + \frac{1-\rho}{(|C|-1)}\sigma_1 D_i)^2 + ((1+\rho+\epsilon)\sigma_1)^2}).$$

Suppose that $\sigma_1 \geq \sigma_0$, the equation holds by setting $\epsilon > 2(1-\rho)D$.

Suppose that $\sigma_0 > \sigma_1$, we have

$$[\rho((1+\rho+\epsilon)^2 + \frac{(1-\rho)^2}{|C|-1}D_i^2)$$
$$+ (1-\rho)|C|(\frac{(1+\rho+\epsilon)(\sigma_0 + \frac{1-\rho}{(|C|-1)}\sigma_1 D_i)^2 + (\frac{1-\rho}{(|C|-1)})((1+\rho+\epsilon)\sigma_1)^2}{(\sigma_0 + \frac{1-\rho}{(|C|-1)}\sigma_1 D_i)^2 + ((1+\rho+\epsilon)\sigma_1)^2})]$$
$$- [\frac{1}{|C|}((1+\rho+\epsilon)^2 + \frac{(1-\rho)^2}{|C|-1}D_i^2)$$
$$+ (1-\frac{1}{|C|})|C|(\frac{(1+\rho+\epsilon)(\sigma_0 + \frac{1-\rho}{(|C|-1)}\sigma_1 D_i)^2 + (\frac{1-\rho}{(|C|-1)})((1+\rho+\epsilon)\sigma_1)^2}{(\sigma_0 + \frac{1-\rho}{(|C|-1)}\sigma_1 D_i)^2 + ((1+\rho+\epsilon)\sigma_1)^2})]$$
$$> [\rho((1+\rho+\epsilon)^2 + \frac{(1-\rho)^2}{|C|-1}D_i^2)$$
$$+ (1-\rho)|C|(1+\rho+\epsilon)] - [\frac{1}{|C|}((1+\rho+\epsilon)^2 + \frac{(1-\rho)^2}{|C|-1}D_i^2) + (1-\frac{1}{|C|})|C|(1+\rho+\epsilon)]$$
$$= (\rho - \frac{1}{|C|})((1+\rho+\epsilon)^2 + \frac{(1-\rho)^2}{|C|-1}D_i^2) - (\rho - \frac{1}{|C|})|C|(1+\rho+\epsilon)]$$
$$= (\rho - \frac{1}{|C|})((1+\rho+\epsilon)^2 + \frac{(1-\rho)^2}{|C|-1}D_i^2 - |C|(1+\rho+\epsilon)].$$

By setting $\rho > \frac{1}{|C|}$ and $\epsilon > |C| - 1$, the above inequality holds. Thus, the theorem follows. $\qquad\square$

### A.9 PROOF OF THEOREM 1

**Theorem 1.** *With nearly orthogonal random features, GCN is provable to distinguish linked and unlinked node pairs if $\frac{1}{8D} > \delta$.*

*Proof.* After one GCN layer,

$$\mathbf{h}_i = \mathbf{x}_i + \frac{1}{D_i}\sum_{v_k \in N(v_i)} \mathbf{x}_k, \text{ and } \mathbf{h}_j = \mathbf{x}_j + \frac{1}{D_j}\sum_{v_l \in N(v_j)} \mathbf{x}_l.$$

If two nodes are linked, we have

$$E[\mathbf{h}_i^\top \mathbf{h}_j]$$
$$= \mathbf{x}_i^\top \mathbf{x}_j + \sum_{v_k \in N(v_i)} \frac{\mathbf{x}_k^\top \mathbf{x}_j}{D_i} + \sum_{v_l \in N(v_j)} \frac{\mathbf{x}_i^\top \mathbf{x}_l}{D_j} + \sum_{v_k \in N(v_i)}\sum_{v_l \in N(v_j)} \frac{\mathbf{x}_k^\top \mathbf{x}_l}{D_i D_j}$$
$$= \frac{1}{D_i} + \frac{1}{D_j} + \mathbf{x}_i^\top \mathbf{x}_j + \sum_{v_k \in N(v_i)\setminus\{v_j\}} \frac{\mathbf{x}_k^\top \mathbf{x}_j}{D_i} + \sum_{v_l \in N(v_j))\setminus\{v_i\}} \frac{\mathbf{x}_i^\top \mathbf{x}_l}{D_j} + \sum_{v_k \in N(v_i)}\sum_{v_l \in N(v_j)} \frac{\mathbf{x}_k^\top \mathbf{x}_l}{D_i D_j}$$
$$\geq \frac{1}{D_i} + \frac{1}{D_j} + \tau_l \frac{D_i - 1}{D_i D_j} - 4\delta.$$

Otherwise,

$$E[\mathbf{h}_i^\top \mathbf{h}_j]$$
$$= \mathbf{x}_i^\top \mathbf{x}_j + \frac{1}{D_i}\sum_{v_k \in N(v_i)} \mathbf{x}_k^\top \mathbf{x}_j + \frac{1}{D_j}\sum_{v_l \in N(v_j)} \mathbf{x}_i^\top \mathbf{x}_l + \frac{1}{D_i D_j}\sum_{v_k \in N(v_i)}\sum_{v_l \in N(v_j)} \mathbf{x}_k^\top \mathbf{x}_l$$
$$\leq \tau_u \frac{1}{D_i} + 4\delta.$$

Based on Proposition 1, we have

$$\frac{1}{D_i} + \frac{1}{D_j} + \tau_l \frac{D_i - 1}{D_i D_j} - 4\delta > \tau_u \frac{1}{D_i} + 4\delta.$$

By transportation,

$$\frac{1 - \tau_u}{D_i} + \frac{1 + \tau_l}{D_j} > 8\delta.$$

By setting $\frac{1}{8D} > \delta$, the theorem follows. □

## A.10 Proof of Theorem 2

**Theorem 2.** *With nearly orthogonal random features, GIN is provable to distinguish linked and unlinked node pairs, if $\epsilon = \frac{D}{2}$ and $\frac{1}{4D^2} > \delta$.*

*Proof.* After one GIN layer,

$$\mathbf{h}_i = (1 + \epsilon)\mathbf{x}_i + \sum_{v_k \in N(v_i)} \mathbf{x}_k, \text{ and } \mathbf{h}_j = (1 + \epsilon)\mathbf{x}_j + \sum_{v_l \in N(v_j)} \mathbf{x}_l,$$

If two nodes are linked, we have

$$E[\mathbf{h}_i^\top \mathbf{h}_j]$$
$$= (1+\epsilon)^2 \mathbf{x}_i^\top \mathbf{x}_j + (1+\epsilon)\left( \sum_{v_k \in N(v_i)} \mathbf{x}_k^\top \mathbf{x}_j + \sum_{v_l \in N(v_j)} \mathbf{x}_i^\top \mathbf{x}_l \right) + \sum_{v_k \in N(v_i)} \sum_{v_l \in N(v_j)} \mathbf{x}_k^\top \mathbf{x}_l$$
$$= 2(1+\epsilon) + (1+\epsilon)^2 \mathbf{x}_i^\top \mathbf{x}_j + (1+\epsilon)\left( \sum_{v_k \in N(v_i)\backslash v_j} \mathbf{x}_k^\top \mathbf{x}_j + \sum_{v_l \in N(v_j)\backslash v_i} \mathbf{x}_i^\top \mathbf{x}_l \right) + \sum_{v_k \in N(v_i)} \sum_{v_l \in N(v_j)} \mathbf{x}_k^\top \mathbf{x}_l$$
$$\geq 2(1+\epsilon) + \tau_l(D_i - 1) - [(1+\epsilon)^2 + (1+\epsilon)(D_i + D_j) + D_i D_j]\delta$$
$$= 2(1+\epsilon) + \tau_l(D_i - 1) - (1 + \epsilon + D_i)(1 + \epsilon + D_j)\delta.$$

Otherwise,

$$E[\mathbf{h}_i^\top \mathbf{h}_j]$$
$$= (1+\epsilon)^2 \mathbf{x}_i^\top \mathbf{x}_j + (1+\epsilon)\left( \sum_{v_k \in N(v_i)} \mathbf{x}_k^\top \mathbf{x}_j + \sum_{v_l \in N(v_j)} \mathbf{x}_i^\top \mathbf{x}_l \right) + \sum_{v_k \in N(v_i)} \sum_{v_l \in N(v_j)} \mathbf{x}_k^\top \mathbf{x}_l$$
$$\leq \tau_u D_i + (1 + \epsilon + D_i)(1 + \epsilon + D_j)\delta.$$

Based on Proposition 1, we have

$$2(1+\epsilon) + \tau_l(D_i - 1) - (1 + \epsilon + D_i)(1 + \epsilon + D_j)\delta > \tau_u D_i + (1 + \epsilon + D_i)(1 + \epsilon + D_j)\delta$$
$$\Rightarrow 2(1+\epsilon) - (1 + \epsilon + D_i)(1 + \epsilon + D_j)\delta > D_i + (1 + \epsilon + D_i)(1 + \epsilon + D_j)\delta$$
$$\Rightarrow 1 + \epsilon > \frac{D_i}{2} + (1 + \epsilon + D_i)(1 + \epsilon + D_j)\delta$$

By setting $\epsilon = \frac{D}{2}$ and $\frac{1}{4D^2} > \delta$, the theorem follows. □

## A.11 Proof of Theorem 3

**Theorem 3.** *With nearly orthogonal random features, GRNN in the following form,*

$$\text{GRNN}^{(k)}(v_i) = \text{MLP}\left((1 + \epsilon)\mathbf{h}_i^{(k-1)} + \sum_{v_j \in N(v_i)} w_j \mathbf{h}_j^{(k-1)}\right),$$

*is provable to distinguish linked and unlinked node pairs, if $\epsilon = \frac{\|\mathbf{w}\|_1}{2}$ and $\frac{4}{13\|\mathbf{w}\|_1^2} > \delta$.*

*Proof.* After one layer GRNN,

$$\mathbf{h}_i = (1 + \epsilon)\mathbf{x}_i + \sum_{v_k \in N(v_i)} w_k \mathbf{x}_k, \text{ and } \mathbf{h}_j = (1 + \epsilon)\mathbf{x}_j + \sum_{v_l \in N(v_j)} w_l \mathbf{x}_l.$$

If two nodes are linked, we have

$$E[\mathbf{h}_i^\top \mathbf{h}_j]$$

$$=(1+\epsilon)^2 \mathbf{x}_i^\top \mathbf{x}_j + (1+\epsilon)(\sum_{v_k \in N(v_i)} w_k \mathbf{x}_k^\top \mathbf{x}_j + \sum_{v_l \in N(v_j)} w_l \mathbf{x}_i^\top \mathbf{x}_l) + \sum_{v_k \in N(v_i)} \sum_{v_l \in N(v_j)} w_k w_l \mathbf{x}_k^\top \mathbf{x}_l$$

$$=2(1+\epsilon) + (1+\epsilon)^2 \mathbf{x}_i^\top \mathbf{x}_j + (1+\epsilon)(\sum_{v_k \in N(v_i) \setminus v_j} w_k \mathbf{x}_k^\top \mathbf{x}_j + \sum_{v_l \in N(v_j) \setminus v_i} w_l \mathbf{x}_i^\top \mathbf{x}_l) + \sum_{v_k \in N(v_i)} \sum_{v_l \in N(v_j)} w_k w_l \mathbf{x}_k^\top \mathbf{x}_l$$

$$\geq 2(1+\epsilon) + \tau_l(\|\mathbf{w}\|_1 - 1) - [(1+\epsilon)^2 + 2(1+\epsilon)\|\mathbf{w}\|_1 + \|\mathbf{w}\|_1^2]\delta$$

$$=2(1+\epsilon) + \tau_l(\|\mathbf{w}\|_1 - 1) - (1+\epsilon+\|\mathbf{w}\|_1)^2\delta.$$

Otherwise,

$$E[\mathbf{h}_i^\top \mathbf{h}_j]$$

$$=(1+\epsilon)^2 \mathbf{x}_i^\top \mathbf{x}_j + (1+\epsilon)(\sum_{v_k \in N(v_i)} \mathbf{x}_k^\top \mathbf{x}_j + \sum_{v_l \in N(v_j)} \mathbf{x}_i^\top \mathbf{x}_l) + \sum_{v_k \in N(v_i)} \sum_{v_l \in N(v_j)} \mathbf{x}_k^\top \mathbf{x}_l$$

$$\leq \tau_u \|\mathbf{w}\|_1 + (1+\epsilon+\|\mathbf{w}\|_1)^2\delta.$$

Based on Proposition 1, we have

$$2(1+\epsilon) + \tau_l(\|\mathbf{w}\|_1 - 1) - (1+\epsilon+\|\mathbf{w}\|_1)^2\delta > \tau_u \|\mathbf{w}\|_1 + (1+\epsilon+\|\mathbf{w}\|_1)^2\delta$$

$$\Rightarrow 2(1+\epsilon) - (1+\epsilon+\|\mathbf{w}\|_1)^2\delta > \|\mathbf{w}\|_1 + (1+\epsilon+\|\mathbf{w}\|_1)^2\delta$$

$$\Rightarrow 1+\epsilon > \frac{\|\mathbf{w}\|_1}{2} + (1+\epsilon+\|\mathbf{w}\|_1)^2\delta.$$

By setting $\epsilon = \frac{\|\mathbf{w}\|_1}{2}$ and $\frac{4}{13\|\mathbf{w}\|_1^2} > \delta$, the theorem follows. $\square$

### A.12 PROOF OF COROLLARY 1

**Corollary 1.** *By uniform sampling $|V|$ embedding vectors as the nearly orthogonal random features, the dimensionality has to be set in the order of $O(\|\mathbf{w}\|_1^4 log(|V|))$ to retain graph reconstructability.*

*Proof.* First, we employ the upper bound of spherical caps (Ball et al., 1997).

$$P(\mathbf{x}_i^T \mathbf{x}_j > \delta) < e^{\frac{-d\delta^2}{2}}$$

$$\Rightarrow 1 - P(\mathbf{x}_i^T \mathbf{x}_j > \delta) \geq 1 - e^{\frac{-d\delta^2}{2}}.$$

By sampling $|V|$ random features for $G$, we have to ensure the orthogonality of the total $\frac{|V|(|V|-1)}{2}$ pairs.

$$(P(\mathbf{x}_i^T \mathbf{x}_j \leq \delta))^{\frac{|V|(|V|-1)}{2}} \geq (1 - e^{\frac{-d\delta^2}{2}})^{\frac{|V|(|V|-1)}{2}}.$$

By Taylor's expansion,

$$(P(\mathbf{x}_i^T \mathbf{x}_j \leq \delta))^{\frac{|V|(|V|-1)}{2}} \geq 1 - \frac{|V|(|V|-1)}{2} e^{\frac{-d\delta^2}{2}}.$$

According to Theorem 3, where $\delta < \frac{4}{13\|\mathbf{w}\|_1^2}$, we obtain

$$(P(\mathbf{x}_i^T \mathbf{x}_j \leq \frac{4}{13\|\mathbf{w}\|_1^2}))^{\frac{|V|(|V|-1)}{2}} \geq 1 - \frac{|V|(|V|-1)}{2} e^{\frac{-8d}{169\|\mathbf{w}\|_1^4}}.$$

Let $p = (P(\mathbf{x}_i^T \mathbf{x}_j \leq \frac{4}{13\|\mathbf{w}\|_1^2}))^{\frac{|V|(|V|-1)}{2}}$, which represents that the initial $|V|$ random features are nearly orthogonal according to $\delta = \frac{4}{13\|\mathbf{w}\|_1^2}$. We have

$$1 - p < \frac{|V|(|V|-1)}{2} e^{\frac{-8d}{169\|\mathbf{w}\|_1^4}}$$

$$\Rightarrow ln(1-p) - ln(\frac{|V|(|V|-1)}{2}) < \frac{-8d}{169\|\mathbf{w}\|_1^4}$$

$$\Rightarrow ln(\frac{|V|(|V|-1)}{2}) - ln(1-p) > \frac{8d}{169\|\mathbf{w}\|_1^4}$$

$$\Rightarrow \frac{169\|\mathbf{w}\|_1^4}{8}(ln(\frac{|V|(|V|-1)}{2}) - ln(1-p)) > d.$$

Since $p$ is closer to 1, the embedding dimension $d$ has to be set in the order of $O(\|\mathbf{w}\|_1^4 log(|V|))$. The corollary follows. $\square$

## A.13 Proof of Proposition 6

**Proposition 6.** *For link prediction, the level of graph reconstructability can be evaluated by the AUC score of the inner product of learned embeddings from GRNN.*

*Proof.* According to Proposition 1, there exists a linear decision boundary to classify the linked and unlinked node pairs. Thus, by sorting the inner product of node embeddings to derive the AUC score, the graph is prefectly reconstructable if AUC is equal to 1. Note that the value of inner prodcut between two nodes indicating the Jaccard similarity, the portion of the common neighborhood over the total neighborhood since the inner product of two none without any shared neighborhood should zero. The proposition follows. □

## A.14 Proof of Proposition 7

**Proposition 7.** *For community detection, the learned embeddings from GRNN approximate the affliction matrix $\mathbf{H}$ via symmetric nonnegative matrix factorization.*

*Proof.* The objective function of community detection (Lu et al., 2020) can be written as follows .

$$\min\|A - \mathbf{H}\mathbf{H}^T\|_F, \text{ s.t. } \mathbf{H} \geq 0 \tag{1}$$

where $\|\|_F$ is the Frobenius norm and $\mathbf{H}$ is the affliattion matrix of each community. $\mathbf{H} \geq 0$ indicate that the elements of $\mathbf{H}$ is nonnegative. Since optimum of graph reconstructability is that the inner product of pairs of node embedding can be obtain classified via a hyperplane as shown in Proposition 1, we can obtained a indicated function (or classifier) to transform the objective of graph reconstructability into Eq. 1. Also, we adopt the `ReLu` activation in Definition 6, thus the learned embeddibg from GNN satisfied $\mathbf{H} \geq 0$. The proposition follows.

□

## A.15 Proof of Corollary 4

**Corollary 4.** *The time complexity of GRNN with NORF is $O(|E|log(|V|) + |V|log^2(|V|))$.*

*Proof.* Since the embedding dimensionality of NORF is $O(log(|V|))$, the dense matrix multiplication of `MLP` requires $O(log^2(|V|))$ of each node. Processing `AGGRGATION` and `COMBINE` requires $O(|E|log(|V|))$ and $O(|V|log(|V|))$. Summing up, the overall time complexity of GRNN with NORF becomes $O(|E|log(|V|) + |V|log^2(|V|))$. □

# B  Additional Experimental Results

## B.1 Setup

Following Kipf & Welling (2017), we adopt a two-layer GNN model, where the projection matrix is initialized as a random unitary matrix. We adopt the skip-gram with negative sampling Hu et al. (2020a) with Adam optimizer. The settings include a learning rate of $10^{-3}$ with a weight decay of $10^{-4}$, dropout rate of 0.2, and a mini-batch size of 32 across all datasets.

## B.2 Running Time

Since NORF is able to reduce the dimensionality of embedding, we further evaluate the running time of GRNN on the synthetic dataset with different graph sizes in Table 6. Since the identity features scheme requires a time complexity of $O(|V|^2)$, it cannot be applied on large graphs, e.g., in our experiments, when the number of nodes is greater than $1,000,000$, we run into the out-of-memory problem. On the other hand, while the contextual features scheme has the best efficiency ($O(|V||C|)$), it does not retain the graph reconstructability on a disassortative graph. In contrast, our NORF can reconstruct all kinds of input graph by ensuring the orthogonality of random features, while significantly reducing the complexity to $O(|V|log(|V|))$.

Table 6: Running Time of GRNN with different features

| # nodes | IF | | | CF | | | NORF | | |
|---|---|---|---|---|---|---|---|---|---|
| | time(sec) | GRR (0.8) | GRR (0.2) | time(sec) | GRR (0.8) | GRR (0.2) | time(sec) | GRR (0.8) | GRR (0.2) |
| 10,000 | 193.23 | 0.9892 | 0.9827 | 9.53 | 0.9731 | 0.6533 | 28.97 | 0.9992 | 0.9921 |
| 100,000 | 27983.66 | 0.9887 | 0.9869 | 112.53 | 0.9802 | 0.6870 | 530.28 | 0.9921 | 0.9896 |
| 1,000,000 | OOM | OOM | OOM | 1437.39 | 0.9723 | 0.6436 | 8921.91 | 0.9871 | 0.9799 |

Table 7: Node classification (ACC.).

| method | Pubmed ($\rho = 0.80$) | | | | Actor (($\rho = 0.22$) | | | |
|---|---|---|---|---|---|---|---|---|
| | IF | CF | NORF | NORF + CF | IF | CF | NORF | NORF + CF |
| GCN | 0.7752 | 0.8648 | 0.7942 | 0.8912 | 0.3244 | 0.3244 | 0.3429 | 0.3544 |
| GAT | 0.7692 | 0.8328 | 0.7880 | 0.8891 | 0.3311 | 0.3221 | 0.3348 | 0.3604 |
| SGC | 0.7741 | 0.8551 | 0.7812 | 0.8998 | 0.3312 | 0.3249 | 0.3519 | 0.3701 |
| GRNN$_{\text{mean}}$ | 0.7541 | 0.8706 | 0.8209 | 0.9110 | 0.3245 | 0.3318 | 0.3550 | 0.3811 |
| GRNN$_{\text{max}}$ | 0.7719 | 0.8719 | 0.8121 | 0.9009 | 0.3503 | 0.3239 | 0.3627 | 0.3721 |
| GRNN$_{\text{attn}}$ | 0.7617 | 0.8921 | 0.8209 | 0.9025 | 0.3519 | 0.3347 | 0.3610 | 0.3777 |

## B.3 NODE CLASSIFICATION

Table 7 summarizes the experimental results of node classification on both assortative (**Pubmed**) (Kipf & Welling, 2017) and disassortative (**Actor**) (Rozemberczki et al., 2021) graphs. We adopt the real node attributes as contextual features. We take $80\%$ of node labels for training and the rest of the datasets for testing. Generally speaking, GRNN achieves the best performance by carefully controlling the weight between neighborhood and central nodes. We observe that, in Pubmed, the contextual features are useful to predict the node labels through the message-passing process in GNN by considering the neighborhood features distribution because the connected node tends to have the same labels on the assortative graph. However, such a procedure would degrade the model performance on the disassortative graph. In contrast, identity features and NORF merely encode the node identity and are not disturbed by irrelevant (and even misleading) contextual information. Note that NORF also outperforms identity features because it requires smaller embedding dimensionality, thus leading to better convergence. Summing up, combining NORF with CF can balance the identity and contextual information in GNN and consistently improve all types of GNN. Among them, GRNN with NORF and CF combined performs the best.

## B.4 MORE REAL-WORLD GRAPHS

Here, we examine the GRNN on more real-world graphs to investigate the correctness of our theoretical framework. Following Liu et al. Liu et al. (2021), we evaluate four GRNN variants against GCN and two GIN variants, coupled with various feature initialization schemes, on both assortative (**Cora** and **Pubmed**) (Kipf & Welling, 2017) and disassortative (**Actor** and **Cornell**) (Rozemberczki et al., 2021) graphs.

## C DISCUSSION

### C.1 GRAPH AUTO-ENCODER.

Generally, graph autoencoder (Wang et al., 2016; 2017) is also a special case of GRNN, which aims to encode the graph (adjacency matrix) in a low dimensional space by minimizing the reconstruction loss, while GRNN is better than graph autoencoder for graph reconstructability. The encoder of graph autoencoder, i.e., the first projection layer, can be regarded as a special case of GRNN with the sum pooling over the randomly initialized features. This is because the input features of the graph autoencoder utilize the row (or column) vector of the adjacency matrix corresponding to each node. Thus, the projection layer of the graph autoencoder can be summarized as $v_i = \sum_{e_{ij} \in E} \mathbf{w}_j^{proj}$, where $\mathbf{w}_j^{proj}$ is the $j$-th row vector of the projection matrix. However, as the graph autoencoder does not employ the self-embedding weight $\epsilon$ for preserving the node identity, it may have difficulty to retain the graph reconstructability. Even if we adopt the self-embedding weight with graph autoencoder, according to Corollary 1, since the aggregation weight $w$ of graph autoencoder is in the order of $O(D)$, where $D$ is the node degree, the embedding dimensionality of the autoencoder is $O(D^4 log(|V|))$

Table 8: Real-World Graph (AUC)

| method | Assortative Graph | | | | | | | |
|---|---|---|---|---|---|---|---|---|
| | Cora ($\rho = 0.81$) | | | | Pubmed ($\rho = 0.80$) | | | |
| | IF | CF | NORF | NORF +CF | IF | CF | NORF | NORF +CF |
| GCN | 0.9931 | 0.9789 | 0.9666 | 0.9712 | 0.9981 | 0.9877 | 0.9632 | 0.9892 |
| GIN | 0.9983 | 0.9894 | 0.9183 | 0.9913 | 0.9987 | 0.9889 | 0.9177 | 0.9959 |
| GAT | 0.9821 | 0.9679 | 0.9712 | 0.9806 | 0.9919 | 0.9766 | 0.9641 | 0.9902, |
| SGC | 0.9791 | 0.9710 | 0.9688 | 0.9744 | 0.9812 | 0.9782 | 0.9429 | 0.9821 |
| GRNN$_{\text{mean}}$ | 0.9925 | 0.9901 | 0.9872 | 0.9799 | 0.9981 | 0.9832 | 0.9854 | 0.9912 |
| GRNN$_{\text{max}}$ | 0.9971 | 0.9843 | 0.9819 | 0.9903 | 0.9902 | 0.9835 | 0.9561 | 0.9913 |
| GRNN$_{\text{attn}}$ | 0.9989 | 0.9788 | 0.9905 | 0.9970 | 0.9986 | 0.9847 | 0.9841 | 0.9903 |
| method | Disassortative Graph | | | | | | | |
| | Actor ($\rho = 0.22$) | | | | Cornell ($\rho = 0.30$) | | | |
| | CF | IF | NORF | NORF +CF | CF | IF | NORF | NORF +CF |
| GCN | 0.9963 | 0.7491 | 0.9670 | 0.9721 | 0.9983 | 0.7325 | 0.9831 | 0.9968 |
| GIN | 0.9984 | 0.7510 | 0.9021 | 0.9112 | 0.9934 | 0.7211 | 0.8814 | 0.8912 |
| GAT | 0.9423 | 0.7001 | 0.9516 | 0.9701 | 0.9891 | 0.7126 | 0.9533 | 0.9778 |
| SGC | 0.9919 | 0.7212 | 0.9560 | 0.9624 | 0.9885 | 0.7360 | 0.9610 | 0.9871 |
| GRNN$_{\text{mean}}$ | 0.9943 | 0.7590 | 0.9863 | 0.9924 | 0.9926 | 0.7390 | 0.9721 | 0.9912 |
| GRNN$_{\text{max}}$ | 0.9865 | 0.7621 | 0.9799 | 0.9829 | 0.9916 | 0.7419 | 0.9801 | 0.9901 |
| GRNN$_{\text{attn}}$ | 0.9966 | 0.7423 | 0.9835 | 0.9912 | 0.9945 | 0.7376 | 0.9757 | 0.9844 |

to retain the graph reconstructability. By contrast, our GRNN only requires $O(log(|V|))$ for the embedding by carefully controlling the self-embedding and aggregation weights, which is much more efficient, especially on a large graph. In addition, the number of training parameters of the graph autoencoder is $O(|V|log(|V|))$, while our GRNN only requires $O(log^2(|V|))$. In this paper, we further explore the theoretical limitation by considering different types of features, i.e., identity and contextual features with assortative and disassortative graphs.

## C.2 POSITIONAL ENCODING

Positional encoding, i.e., encoding the global position of phones in audios (Park et al., 2021), pixels in images (Dosovitskiy et al., 2020), and words in texts (Wang & Chen, 2020), plays a crucial role in many prominent neural networks. For GNNs, encoding the position of nodes is very challenging since there exists no canonical positioning of nodes in graphs. Therefore, nodes in a graph can be assigned one-hot encoding in accordance with the node index (Chen et al., 2022), enabling GNNs to derive more expressive power than the 1-WL test (Murphy et al., 2019). Sato et al. (Sato et al., 2021) demonstrate that GNNs become more powerful by adding a random feature to each node, which obtains almost optimal polynomial-time approximation algorithms for the minimum dominating set and the maximum matching problem. However, these works mainly focus on analyzing the expressive power of GNNs from a global view. Our theoretical results manifest that identity and (nearly) orthogonal random features can effectively preserve topological information from the graph.

## C.3 SKIP-GRAM EMBEDDING

The importance of reconstructability for skip-gram-based embedding is revealed by a recent work (Chanpuriya et al., 2021), which investigates whether embedding methods learned by the skip-gram objective can reconstruct the original graph. The authors subsequently employ the embedding algorithm to solve several fundamental network mining tasks, including common edges, degree sequences, triangle counts, and community structure. This is because the skip-gram-based embedding schemes could be summarized in various forms of matrix factorization over the Pairwise Mutual Information (PMI) Matrix (Qiu et al., 2018). Nevertheless, they do not consider the factor of node features (e.g., attributes or identity encoding) and the graph topology (e.g., graph degree and homophily). In addition, we theoretically analyze the dimensionality of embedding and reduce the complexity from $O(|V|)$ to $O(log(|V|))$, while retaining the reconstructability by properly designing the GNN and initial features.