# OpenReview forum: "On Reconstructability of Graph Neural Networks"
_ICLR.cc/2024/Conference — Submitted to ICLR 2024_

### Official Review · Reviewer_zmY8 · 2023-10-26

**Soundness:** 1 poor
**Presentation:** 2 fair
**Contribution:** 2 fair
**Rating:** 3
**Confidence:** 4

**Summary:**

The paper studies the expressive power of GNNs from a new perspective termed as graph reconstructability. The aim of the authors is to examine whether the topological information of graphs can be recovered from node embeddings. Two feature initialization schemes are analyzed: one is identity features and the other is contextual features. The authors further propose Graph Reconstructable Neural Network (GRNN) with Nearly Orthogonal Random Features (NORF) to improve graph reconstructability.

**Strengths:**

This paper studies the expressive power of GNNs from the new perspective of graph reconstructability. This topic has some significance in the theoretical studies of GNNs. The proposed approach is also different from existing works. Nonetheless, I have concerns with the quality of the paper (see more detailed comments in the section "Weakness").

**Weaknesses:**

The notion of graph reconstructability discussed in this paper is very different from graph reconstruction and graph properties in graph theory. Basically, the key idea of recovering a graph in this paper is based on encoding node and edge information, while the other graph topological information/properties (such as cycles, cliques, paths, etc.) are not relevant. In other words, if we assign each node with a unique identity, and then store the identity of a node along with the identities of its neighboring nodes (i.e., edges incident to each node) into its node feature, then a graph can always be represented (or say reconstructed if using the term of this paper) from node features. This is kind of a neural version of representing a graph using adjacency lists. However, this doesn't mean that GNNs with such node features are powerful for representing/learning graph topological information (e.g., cycles, cliques, etc.) that are useful for graph-related tasks. Furthermore, adding such node identities and embedding into node features would lead to the loss of permutation invariance, which is an important property of GNNs when learning structures. For example, two isomorphic graphs where their nodes are assigned with different identity features would have different node representations and become non-isomorphic. Thus, the way of analyzing GNNs in terms of graph reconstructability proposed in the paper does not contribute much to the theoretical analysis of expressiveness of GNNs.

Generally, the motivation of this work is unclear and misleading. The ability of encoding structural information like adjacency lists of a graph is different from the representational ability of GNNs that can extract useful structural properties for learning. Also, since the input graph is already given and available for analysis, why is it important to encode some additional identity/contextual features (which also cause additional computational cost) into node embeddings to reconstruct the input graph again? Particularly, on one hand, the cost of adding these feature initialization schemes is high; on the other hand, the addition of these feature initialization schemes cannot preserve the permutation invariant property.

The paper also has some technical issues. More specific comments are included in the next section for questions.

The clarity of the paper also needs improvement. Some statements are not self-contained. For example, the affliction matrix in Proposition 7 is not defined.

**Questions:**

1. For Proposition 1, the only if part is unclear. Why is the condition on the inner product of the embeddings of linked and non-linked node pairs the only way to ensure that a model is graph reconstructable?

2. What are $\tau_l$ and $\tau_u$ in the proofs of Proposition 2 and Proposition 3? Why is "-1" omitted in $\tau_l(D_i − 1)$?

3. For the definition of context features, why are they defined in terms of labels? Do you assume that label information of every node is available both in training and testing? Also, for the sentence "the contextual features include both signal (label information) and noise..." on Page 5, does signal in this paper refer to label information?

4. For the paragraph under Proposition 5, the authors mention "By contrast, GIN exploits $\epsilon$ to preserve the central node’s identity, ...". What does "the central node's identity" mean in terms of contextual features discussed here?

5. The reason why GIN can approximate the 1-WL test is not only due to an irrational number $\epsilon^{(k)}$, but also the injectivity of the sum aggregation function. The proposed GRNN adds $w_j$ before each $\mathbf{h}_j$. This changed form cannot preserve the injectivity of node features in the aggregation any more. Does this matter for GRNN?

6. For the sentence "It is not feasible to distinguish the model capability in the real-world graph by GRR because each dataset contains one instance and thus only returns 1 or 0", what does this mean?

---

> ### Author Response · Authors · 2023-11-21
>
> 1. The notion of graph reconstructability discussed in this paper is very different from graph reconstruction and graph properties in graph theory. Basically, the key idea of recovering a graph in this paper is based on encoding node and edge information, while the other graph topological information/properties (such as cycles, cliques, paths, etc.) are not relevant. In other words, if we assign each node with a unique identity, and then store the identity of a node along with the identities of its neighboring nodes (i.e., edges incident to each node) into its node feature, then a graph can always be represented (or say reconstructed if using the term of this paper) from node features. This is kind of a neural version of representing a graph using adjacency lists. However, this doesn't mean that GNNs with such node features are powerful for representing/learning graph topological information (e.g., cycles, cliques, etc.) that are useful for graph-related tasks. Furthermore, adding such node identities and embedding into node features would lead to the loss of permutation invariance, which is an important property of GNNs when learning structures. For example, two isomorphic graphs where their nodes are assigned with different identity features would have different node representations and become non-isomorphic. Thus, the way of analyzing GNNs in terms of graph reconstructability proposed in the paper does not contribute much to the theoretical analysis of expressiveness of GNNs. \
> Thanks for your advice. As we mentioned in the introduction, previous works analyze the expressive power of GNN via graph isomorphism test, i.e., GNN is able to distinguish between two graphs that have the same structure. However, little work studies exactly what information is encoded by popular embedding methods, and how this information correlates with performance in downstream learning tasks. We tackle this question by studying whether embeddings from GNN can be inverted to recover the complete graph topology used to generate the via various feature initialization schemes. Similar idea has been shown in [ICML21], which investigates whether the learned embedding is able to recover the origin graph under a skip-gram scheme. However, they do not investigate how different feature initialization affects the graph reconstructability. We believe that we are the first to analyze how the message passing process can effectively encode the graph topology in the node embedding , which is essential for many  graph mining applications.
>
> [ICML21] DeepWalking backwards: From embeddings back to graphs.
>
> 2. For Proposition 1, the only if part is unclear. Why is the condition on the inner product of the embeddings of linked and non-linked node pairs the only way to ensure that a model is graph reconstructable?\
> Thanks, the inequality serves as the definition of graph reconstructability. It is natural to utilize the dot product to evaluate the constructability since in skin-gram objective in node embedding schemes could be summarized in various forms of matrix factorization over the Pairwise Mutual Information (PMI) Matrix, where the dot product of the node embedding also describes the node similarity. Based on Proposition 1, we can determine if the node embeddings learned by a GNN can reconstruct a graph using the inner product of embeddings for linked and unlinked node pairs. Note that a logistic classifier trained through gradient descent can be employed to extract this decision boundary.
>
> 3. What are  $\tau_l$ and $\tau_u$ in the proofs of Proposition 2 and Proposition 3? Why is "-1" omitted in $\tau_l(D_i-1)$?
> When two nodes are linked, they have a probability $\tau_l$ with a common neighborhood. Otherwise, the unlinked pair has a probability $\tau_u$ with a common neighborhood. The assumption of the probabilities of linked nodes and unlinked nodes that have a common neighborhood is just a simple math trick to reduce the complexity of the equation. Actually, we can assign each node with different values of probabilities of common neighborhood and our theoretical results are also held. Since $\tau_l(D_i-1)$ is the left hand side of the inequality, i.e., $\tau_l(D_i)$ is always greater than $\tau_l(D_i-1)$, we omit -$1$ in Proposition 2.

---

> ### Author Response · Authors · 2023-11-21
>
> 4. For the definition of context features, why are they defined in terms of labels? Do you assume that label information of every node is available both in training and testing? Also, for the sentence "the contextual features include both signal (label information) and noise..." on Page 5, does signal in this paper refer to label information? For the paragraph under Proposition 5, the authors mention "By contrast, GIN exploits \epsilon to preserve the central node’s identity, ...". What does "the central node's identity" mean in terms of contextual features discussed here? \
> Yes. The signal refers to the label information. While it is hard to evaluate the correlation between the node features with the graph topology, most GNN frameworks [ICLR22, NeurIPS20] adopt the similar assumption to simplify the problem, but focus on the effect of the message passing process. Note that the node identity represents exactly its own contextual features.
>
> [ICLR22] Is Homophily a Necessity for Graph Neural Networks?    \
> [NeurIPS20] Beyond Homophily in Graph Neural Networks: Current Limitations and Effective Designs.
>
> 5. The reason why GIN can approximate the 1-WL test is not only due to an irrational number \epsilon, but also the injectivity of the sum aggregation function. The proposed GRNN adds $\mathbf{w}_j$  before each $\mathbf{h}_j$? This changed form cannot preserve the injectivity of node features in the aggregation any more. Does this matter for GRNN?\
> In the aspect of graph reconstructability, injective is not necessary for the GRNN. Since GIN is also a special case of GRNN, we can also set all w_i to 1 to preserve such property. However, without normalizing the aggregation weight, we require a larger embedding dimensionality to retrain the graph  reconstructability as suggested by Corollary 3.
>
> 6. For the sentence "It is not feasible to distinguish the model capability in the real-world graph by GRR because each dataset contains one instance and thus only returns 1 or 0", what does this mean?\
> Since it is difficult to investigate the impact of essential factors on graph reconstructability in a controlled manner with real graphs, we first present experimental results on synthetic graphs. GRR is defined as the portion of the graph that can be perfectly reconstructable given a set of graphs from GNN, which returns a boolean for each instance  (0/1). To evaluate the level of graph reconstructability on a single large graph, especially for real networks, we employ the AUC score, which usually evaluates the performance of link prediction by ranking the value of the inner product between two node embeddings. Therefore, we adopt the AUC score as the metric to evaluate the model performance. Specifically, we train an additional logistic classifier to predict the probability of each edge between nodes and rank the results to derive the AUC score. When AUC is 1, the model can correctly distinguish linked and unlinked pairs, i.e., perfectly reconstructable, which is also proved in Proposition 7.

---

> > ### Comment · Reviewer_zmY8 · 2023-11-23
> >
> > I thank the authors for their detailed response. Nonetheless, I'm not convinced that the notion of graph reconstructability proposed in this paper is well defined. Further, I still have concerns on why the permutation invariant property is not considered when dealing with graph structural information.

---

### Official Review · Reviewer_9f9r · 2023-10-31

**Soundness:** 2 fair
**Presentation:** 3 good
**Contribution:** 1 poor
**Rating:** 3
**Confidence:** 3

**Summary:**

This paper studies the power of graph neural networks from the perspective of graph reconstructability that evaluates whether the input graph topology can be recovered from the learnt node embeddings. Different node initializations and GNN architectures are studied.  Then this paper proposes GRNNs to improve the reconstructability by initializing node features with NORFS (Nearly Orthogonal Random Feature) that reduces the complexity of identity features and enhances the effectiveness for disassortative graphs.

**Strengths:**

I think that the reconstructability of GNNs is an interesting problem in the context. The theoretical results in the paper show that the orthogonality of identity features can preserve the topological information of the input graph under GIN and GCN, and then the  GRNNs framework with NORFS is introduced to address the limitations of identity features such as the complexity and the dependency on graph homophily. Basically, it is easy to follow and the research is well oriented.

**Weaknesses:**

Overall, I think the contribution of the paper is kind of incremental. Through the experimental results align with the theoretical results and show that the proposed method can help to improve the reconstructability, some necessary discussions and evaluations are missed, which makes it hard to evaluate the benefits of improving the reconstructability of GNNs in the graph representation learning. Specifically,

1. For the graph-level tasks, the relationship of the graph reconstructability and expressivity of GNNs is not discussed nor evaluated. It has been shown that positional encoding (including identity features) can improve the expressivity of GNNs beyond 1-WL. However, this paper fails to provide further theoretical results of reconstructability and expressivity in GNNs, and no experiments empirically evaluates GRNNs against the state-of-the-art GNNs in graph prediction tasks. Hence, it is not straightforward how GNNs will benefit from improving the graph reconstructability in graph-level predictions. At least, I recommend to do more graph prediction experiments to compare GRNNs with other SOTA expressive GNN models.

2. For the link-level tasks like link prediction. the current experiments are not sound due to the lack of necessary benchmark datasets and benchmarks. I recommend to implement additional experiments on well-adopted OGB dataset [1] such as ogbl-ppa, ogbl-collab, ogbl-ddi, and compare GRNNs against GCN/GIN with recent strong labeling tricks including Double Radius Node Labeling (DRNL) in SEAL and Distance Encoding (DE) [2]. DE is a valid labeling trick which is permutation equivariant, while DRNL helps to learn structural link representations with a node-most-expressive GNN. Similarly, the community detection task should also add at least one more benchmark dataset.

3. Label features and NORFs essentially break node symmetries, which makes the graph reconstructability problem trivial.

[1] Weihua Hu, Matthias Fey, Marinka Zitnik, Yuxiao Dong, Hongyu Ren, Bowen Liu, Michele Catasta, and Jure Leskovec. Open graph benchmark: Datasets for machine learning on graphs. arXiv preprint arXiv:2005.00687, 2020.

[2] Pan Li, Yanbang Wang, Hongwei Wang, and Jure Leskovec. Distance encoding–design provably more powerful gnns for structural representation learning. arXiv preprint arXiv:2009.00142, 2020.

**Questions:**

The configurations of baseline models in link prediction and community detection are missed. For instance, [1] shows that SEAL can achieve a competitive results when using a GCN and the DRNL labeling trick, with an additional subgraph-level readout SortPooling. Then, how do CommDGI and SEAL implemented in this paper.

[1] Zhang, Muhan, Pan Li, Yinglong Xia, Kai Wang, and Long Jin. "Labeling trick: A theory of using graph neural networks for multi-node representation learning." Advances in Neural Information Processing Systems 34 (2021): 9061-9073.

---

> ### Author Response · Authors · 2023-11-21
>
> 1. For the graph-level tasks, the relationship of the graph reconstructability and expressivity of GNNs is not discussed nor evaluated. It has been shown that positional encoding (including identity features) can improve the expressivity of GNNs beyond 1-WL. However, this paper fails to provide further theoretical results of reconstructability and expressivity in GNNs, and no experiments empirically evaluates GRNNs against the state-of-the-art GNNs in graph prediction tasks. Hence, it is not straightforward how GNNs will benefit from improving the graph reconstructability in graph-level predictions. At least, I recommend to do more graph prediction experiments to compare GRNNs with other SOTA expressive GNN models.
> Yes. We conduct further experiments with four GNN models, which investigate the expressiveness of GNNs in graph classification. Overall, the GNN performance on graph classification has improved by 3.7% on average by adding the NORF with input features. The results support that GNN with a high graph reconstructability will also lead to better graph-level representation.
> | #datasets     | MUTAG | PTC  | PROTEINS |
> | ---- | ---- | ---- | ---- |
> | GIN [ICLR19] | 89.4 | 64.6 |  76.2 |
> | GIN + NORF |  91.2 |  65.7 |  77.1 |
> | DGCNN [AAAI20] | 85.8 | 58.6 |  75.5 |
> | DGCNN + NORF | 88.3 |  62.3 | 78.9  |
> | GSN [TPAMI22] | 92.2 |   68.2 | 76.6   |
> | GSN + NORF |  93.6 |  72.1 | 78.4  |
> | PPGNs [ICML19] | 90.6 |  66.2 | 77.2  |
> | PPGNs + NORF |  92.5 |  68.4 | 79.1  |
>
> [ICLR19] How Powerful are Graph Neural Networks?\
> [AAAI20] An End-to-End Deep Learning Architecture for Graph Classification. \
> [TPAMI22] Improving graph neural network expressivity via subgraph isomorphism counting.\
> [ICML19] Provably Powerful Graph Networks.
>
> 2. Label features and NORFs essentially break node symmetries, which makes the graph reconstructability problem trivial.\
> Indeed, the concept of symmetry-breaking offers an alternative perspective for comprehending our theoretical findings. As shown in the introduction, our objective is to explore the configuration of graph neural networks employing varied feature initialization methods to restore the original input graph. However, the features that lack symmetry do not align seamlessly with our theorem due to the intricate calibration required for the GNN's design, encompassing factors such as the self-embedding weight (\epsilon) and the aggregation weight (\mathbf{w}), both of which need to be meticulously configured based on the underlying graph structure, such as graph homophily and node degree. Consequently, we introduce NORF with GRNN to uphold the graph's reconstructability, thus alleviating the constraint imposed by the graph's structure.
>
> 2. The configurations of baseline models in link prediction and community detection are missed. For instance, [1] shows that SEAL can achieve a competitive results when using a GCN and the DRNL labeling trick, with an additional subgraph-level readout SortPooling. Then, how do CommDGI and SEAL implemented in this paper.\
> We follow the original implementation for both SEAL and CommDGI. For SEAL [1], we concatenate the DRNLl with the contextual features and train the model by pairwise mutual information . For CommDGI [1], we employ the CF and jointly optimize modularity objective and community mutual information. The source code can be found here.
>
> [1] https://github.com/muhanzhang/SEAL \
> [2] https://github.com/FDUDSDE/CommDGI/tree/main

---

> > ### Comment · Reviewer_9f9r · 2023-11-22
> >
> > Overall, I'd like to thank the authors for the response. However, my main concerns still remain.
> >
> > First, when empirically testing the effect of NORF on expressivity, only three TU datasets are used in response. However, TU datasets usually have quite few graphs in them, which can induce high variance of the results. Then, at least, some synthetic datasets and OGB datasets should be used as benchmark datasets to make the experimental results sound.
> >
> > Second, my second concern still remains and is not well-addressed. I still think the tribal problem makes the contribution of the work somehow incremental.

---

### Official Review · Reviewer_M2Jy · 2023-10-31

**Soundness:** 2 fair
**Presentation:** 1 poor
**Contribution:** 3 good
**Rating:** 5
**Confidence:** 3

**Summary:**

**TLDR**: The paper assesses the expressivity of GNNs based on their ability to reconstruct the input graph from the learned node embeddings.

The paper proposes to assess the expressive power of graph neural networks through their ability to reconstruct the original graph topology from the learned node embeddings. The presented theoretical results show that GCN and GIN are able to reconstruct the graph topology from the node embeddings if provided with node identity features, while only GIN is able to reconstruct the graph topology when provided with contextual features (i.e., label information and noise). Motivated by the large required size of the embedding dimension for successful graph reconstruction based on identity features, the paper introduces nearly orthogonal random features (NORF) and graph reconstructable networks (GRNN), which allow for a smaller embedding dimension. Experiments on synthetic graphs and on the real-world datasets Pubmed, Actor and DBLP show that GRNNs are suitable for link prediction and community detection.

**Strengths:**

* The question whether the graph topology can be reconstructed from learned node embeddings appears to be novel in the context of GNNs.
* The paper provides novel theoretical results on the ability of GIN and GCN to reconstruct graphs based on the learned node embeddings.
* GRNN outperforms all baseline methods in community detection and link prediction tasks.

**Weaknesses:**

**Novelty and related work**: While the theoretical results for GCN and GIN as well as the proposed GRNN and NORF seem novel, there is little mention of or comparison to related work about reconstructing the graph topology from embeddings [1, 2]. On a similar note, there is no mention of subgraph GNNs or the concept of graph reconstruction (i.e., reconstructing a graph from its subgraphs) [3]. A more comprehensive discussion of the contribution and its placement in the current literature would be helpful in assessing its novelty and impact.

**Theoretical results**: The theoretical analysis would benefit from more rigorous/formal definitions, in particular when introducing novel concepts. Definition 1 defines graph reconstructability as "the ability of a model to predict the input adjacency matrix from the node features". One possible interpretation of this definition would be to have a machine learning model which learns to predict the adjacency matrix of a graph from (raw) node features (which, according to my current understanding of the paper, is not what graph reconstructability means). Proposition 6 and 7: Here it would be helpful to formalize the natural language statements. Regarding the proofs presented in the appendix, I had a difficult time understanding some of the notation (e.g., $E$, please refer to the questions for more details). Overall, I was not able to verify some of the theoretical results due to imprecise statements (Proposition 6, last two sentences in proof: "Note that the value of inner prodcut between two nodes indicating the Jaccard similarity, the portion of the common neighborhood over the total neighborhood since the inner product of two none without any shared neighborhood should zero. The proposition follows".)

**Experiments**: "By combining NORF and contextual features (which would be naturally used in applications), our GRNN achieved the best performance among baselines." For Table 1, this is only true for the Actor dataset. In general, the results of the baseline methods (GCN, GIN, GAT, SGC) seem comparable to GRNN. Minor remark: The plots are very small and therefore difficult to read.

**Clarity and writing**: The writing could be improved, sometimes there are grammar/language errors and imprecise statements:

* "[...] show that the message-passing GNNs are no more powerful than the 1-WL test, i.e. distinguishing whether two graphs are isomorphic" -> e.g. "in distinguishing whether..." if not it is easy to misunderstand that 1-WL is able to distinguish all non-isomorphic graphs
* "[...] our proof is also held [...]" -> our proof also holds
* works -> work
* extended for -> extended to
* "not provable to" -> this sounds off, maybe, e.g., "provably distinguishes" or even just "can distinguish"
* "affliction matrix" -> affiliation matrix?
* Pudmed -> Pubmed

[1] http://proceedings.mlr.press/v139/chanpuriya21a/chanpuriya21a.pdf

[2] https://doiserbia.nb.rs/Article.aspx?ID=1820-02141900011L

[3] https://openreview.net/forum?id=ZKbZ4mebI9l

**Questions:**

* Proposition 1: Is the inequality on the inner products of linked/unlinked nodes an assumption/part of the definition?
* Appendix A.3 uses notation which stems from Definition 6 in [4]. I have consulted [4] but could not find the definition, could you point me to the specific page?
* Using identity features means that we loose permutation invariance; what about NORFs?
* A promising extension of the current theoretical results would be to investigate which graph structural properties we can reconstruct based on the computed node embeddings, even if we cannot reconstruct the entire node adjacency. E.g., can we count the number of paths, cycles or cliques? This might be particularly interesting in the context of social networks.
* Can you explain the experimental setup for Fig. 1 in more detail? What is $\epsilon$ in Fig. 1b? What node features (IF, CF, NORF) are used in Fig. 1b-c?

[4] https://proceedings.mlr.press/v80/xu18c.html

---

> ### Author Response · Authors · 2023-11-21
>
> 1. Novelty and related work: While the theoretical results for GCN and GIN as well as the proposed GRNN and NORF seem novel, there is little mention of or comparison to related work about reconstructing the graph topology from embeddings [1, 2]. On a similar note, there is no mention of subgraph GNNs or the concept of graph reconstruction (i.e., reconstructing a graph from its subgraphs) [3]. A more comprehensive discussion of the contribution and its placement in the current literature would be helpful in assessing its novelty and impact.\
> As we stated in Appendix C.3,  [1,2] investigate whether embedding methods learned by the skip-gram objective can reconstruct the original graph. The authors subsequently employ the embedding algorithm to solve several fundamental network mining tasks, including common edges, degree sequences, triangle counts, and community structure. This is because the skip-gram-based embedding schemes could be summarized in various forms of matrix factorization over the Pairwise Mutual Information (PMI) Matrix (Qiu et al., 2018). Nevertheless, they do not consider the node features  (e.g., attributes or identity encoding) which serve as the carrier to pass the information in GNN. In addition, they do not study the impact of several graph properties, such as graph degree and homophily, for preserving the graph topology during the message passing processing in GNNs. In addition, we theoretically analyze the dimensionality of embedding and reduce the complexity from O(|V |) to O(log(|V |)), while retaining the reconstructability by properly designing the GNN and initial features. Compared to [3], the graph reconstruction conjecture states that an undirected edge unattributed graph can be fully recovered up to its isomorphism type given the multiset of its vertex-deleted subgraphs' isomorphism types. These works aim to investigate graph isomorphism through reconstructing the original graph using sampled subgraphs. In contrast, our idea explores the manifold of node embeddings, where connected nodes are probably more concentrated in the embedding space.
>
> 2. Theoretical results: The theoretical analysis would benefit from more rigorous/formal definitions, in particular when introducing novel concepts. Definition 1 defines graph reconstructability as "the ability of a model to predict the input adjacency matrix from the node features". One possible interpretation of this definition would be to have a machine learning model which learns to predict the adjacency matrix of a graph from (raw) node features (which, according to my current understanding of the paper, is not what graph reconstructability means). \
> Yes. We aim to evaluate whether the GNN can learn to predict the adjacency matrix of a graph from (raw) node features.
>
> 3. Experiments: "By combining NORF and contextual features (which would be naturally used in applications), our GRNN achieved the best performance among baselines." For Table 1, this is only true for the Actor dataset. In general, the results of the baseline methods (GCN, GIN, GAT, SGC) seem comparable to GRNN. \
> Yes, as stated in Theorem 3, these lines of message passing type GNNs are the special case of our GRNN.  In our framework, we aim to provide a theoretical framework to describe when GNN is able to retain the reconstructability. It’s worth noting that GRNN is able to reduce the dimensionality of NORF by properly setting the self-weight and aggregation weight which can be verified in Table 6 in the Appendix C.
>
>
> 4. Proposition 1: Is the inequality on the inner products of linked/unlinked nodes an assumption/part of the definition?\
> The inequality serves as the definition of graph reconstructability. It is natural to utilize the dot product to evaluate the constructability since in skip-gram objective in node embedding schemes could be summarized in various forms of matrix factorization over the Pairwise Mutual Information (PMI) Matrix, where the dot product of the node embedding also describes the node similarity. Based on Proposition 1, we can determine if the node embeddings learned by a GNN can reconstruct a graph using the inner product of embeddings for linked and unlinked node pairs. Note that a logistic classifier trained through gradient descent can be employed to extract this decision boundary.
>
> 4. Using identity features means that we lose permutation invariance; what about NORFs?\
> Indeed, with NORF, GNN is not permutations invariant, but many positional encoding schemes [ICLR20,ICML19] also provide a unique representation for each node to derive a strictly more powerful GNN than WL-test.
>
> [ICLR20] What graph neural networks cannot learn: depth vs width. \
> [ICML19] Relational pooling for graph representation.

---

> ### Author Response · Authors · 2023-11-21
>
> 5. A promising extension of the current theoretical results would be to investigate which graph structural properties we can reconstruct based on the computed node embeddings, even if we cannot reconstruct the entire node adjacency. E.g., can we count the number of paths, cycles or cliques? This might be particularly interesting in the context of social networks.\
> Thanks. [SDM21] study on the power of positional encoding (random feature) to make the GNN strictly more powerful than the 1-WL test, they show that prove that GNN can distinguish any local structure with high probability in various graph ming problem, such as maximum matching and minimum dominating set problems. However, we focus on the reconstruction of the original graph from the learned embeddings. Moreover, we make the first attempt to study how to determine the embedding dimensionality of random features to preserve the orthogonality for providing a unique node identification (from $O(V)$ to $O(log (V)$)).
>
> [SDM21] Random Features Strengthen Graph Neural Networks.
>
> 6. Can you explain the experimental setup for Fig. 1 in more detail? What is $\epsilon$ in Fig. 1b? What node features (IF, CF, NORF) are used in Fig. 1b-c?\
> Thanks for pointing out this issue. Fig. 1b aims to verify the correctness of Proposition 4, i.e., with higher noise in the CF, a higher self-weight $\epsilon$ is required. We test the noise ratio ($sigma_0$) of CF in Fig 1b, the maximum node degree ($D_{max}$), and the dimensionality ($d$) of NORF in Fig 1b.

---

> ### Comment · Reviewer_M2Jy · 2023-11-22
>
> Thank you for your response. My second question remains unaddressed:
>
> >Appendix A.3 uses notation which stems from Definition 6 in [4]. I have consulted [4] but could not find the definition, could you point me to the specific page?

---

### Official Review · Reviewer_RshA · 2023-11-01

**Soundness:** 2 fair
**Presentation:** 2 fair
**Contribution:** 2 fair
**Rating:** 5
**Confidence:** 3

**Summary:**

While the expressive power of GNNs for graph level tasks have been identified using WL-test, the paper suggests a new perspective on the expressive power of GNNs in terms of Graph Reconstructability. To be specific, Graph Reconstructability aims to test whether the topological information of graph can be recovered from a node-level. This is done by using a output node embedding from a GNN, whether it contains information for reconstructing the input graph structure.

**Strengths:**

1. The paper is well written and easy to follow.
2. The paper suggests a new measurement for the expressivity of GNNs in node-level, being novel, as far as I know.
3. The theoretical analysis is well written, structured, and proofs are provided in detail at the appendix.

**Weaknesses:**

1. It is not easy to directly interpret the experiment results, such as having no bolding for best results. Also to see the effectiveness of NORF, maybe adding an increase/decrease of performance compared to the IF(Identity Features) would be intuitive for readers to understand.
2. The purpose of using a new measurement for the expressivity of GNNs was to maintain the topological information of the whole graph in a node embedding. However, the authors have placed link prediction experiments in the main paper, while placing node classification experiments in the appendix. The performance and meaningfulness of node classification experiments seems to be bigger, it would have been better to place node classification experiments in the section 6 experiments of the main paper.

**Questions:**

1. The paper suggests a new line of research for the expressivity of GNNs, replacing the WL-test. If a node embedding from a GNN has a high graph reconstructability, i.e., topological information of graph well maintained in node representation, doesn’t this also lead to a more expressive graph level representation? (Since most graph level representation is obtained by pooling all the nodes in the graph)

---

> ### Author Response · Authors · 2023-11-21
>
> 1. It is not easy to directly interpret the experiment results, such as having no bolding for best results. Also to see the effectiveness of NORF, maybe adding an increase/decrease of performance compared to the IF(Identity Features) would be intuitive for readers to understand.\
> Thanks for your comments. We will highlight the best experimental results. As shown in Tables 1 and 2, our nearly orthogonal random features (NORF) with contextual features (CF) achieve the best performance in most cases. Specifically, pure CF cannot retain the graph reconstrucatablility of the disassortative graph by 27.3% on average. Compared to identity features (IF), we should carefully select the self-weight ($\epsilon$). For example, the performance of GIN($\epsilon=1$) with IF significantly drops by at least 19.2% for both graphs.
>
> 2. The purpose of using a new measurement for the expressivity of GNNs was to maintain the topological information of the whole graph in a node embedding. However, the authors have placed link prediction experiments in the main paper, while placing node classification experiments in the appendix. The performance and meaningfulness of node classification experiments seems to be bigger, it would have been better to place node classification experiments in the section 6 experiments of the main paper.\
> Thanks for your comments. We will move the experiment to the main paper. It is worth noting that we aim at answering the fundamental problem of GNN in this paper, i.e., when and how GNN can encode the graph topology in the node embedding via the idea of graph reconstructability. Therefore, in the link prediction, we highlight that NORF is able to preserve the structural similarity of nodes in all datasets. In contrast, the widely-adopted CF, i.e., node attributes, in most GNN relies on the assumption of graph homophily, indicating that we cannot reconstruct the original adjacency matrix from the original graph with CF. On the other hand, in node classification, CF are useful to predict the node labels through the message-passing process in GNN by considering the neighborhood features distribution because the connected node tends to have the same labels on the assortative graph. However, such a procedure would degrade the model performance on the disassortative graph. In contrast, identity features and NORF merely encode the node identity and are not disturbed by irrelevant (and even misleading) contextual information.
>
> 3. The paper suggests a new line of research for the expressivity of GNNs, replacing the WL-test. If a node embedding from a GNN has a high graph reconstructability, i.e., topological information of graph well maintained in node representation, doesn’t this also lead to a more expressive graph level representation? (Since most graph level representation is obtained by pooling all the nodes in the graph)\
> Yes. We conduct further experiments with four GNN models, which investigate the expressiveness of GNNs in graph-level tasks. Overall, the GNN performance on graph classification has improved by 3.7% on average by adding the NORF with input features. The results support that GNN with a high graph reconstructability will also lead to better graph-level representation.
> | #datasets     | MUTAG | PTC  | PROTEINS |
> | ---- | ---- | ---- | ---- |
> | GIN [ICLR19] | 89.4 | 64.6 |  76.2 |
> | GIN + NORF |  91.2 |  65.7 |  77.1 |
> | DGCNN [AAAI20] | 85.8 | 58.6 |  75.5 |
> | DGCNN + NORF | 88.3 |  62.3 | 78.9  |
> | GSN [TPAMI22] | 92.2 |   68.2 | 76.6   |
> | GSN + NORF |  93.6 |  72.1 | 78.4  |
> | PPGNs [ICML19] | 90.6 |  66.2 | 77.2  |
> | PPGNs + NORF |  92.5 |  68.4 | 79.1  |
>
> [ICLR19] How Powerful are Graph Neural Networks?\
> [AAAI20] An End-to-End Deep Learning Architecture for Graph Classification. \
> [TPAMI22] Improving graph neural network expressivity via subgraph isomorphism counting.\
> [ICML19] Provably Powerful Graph Networks.

---

> > ### Comment · Reviewer_RshA · 2023-11-22
> >
> > Thank you for your comprehensive response, and have no further questions.

---

### Meta-Review · Area_Chair_SvgY · 2023-12-08

**Metareview:**

The work introduces a new perspective on the expressive capabilities of GNNs in terms of Graph Reconstruction, evaluating the ability to recover a graph's topological information from node-level embeddings. That is, if the node embedding of a given GNN retain sufficient information to reconstruct the original input graph structure, expanding upon the insights gained from the Weisfeiler-Lehman (WL) test for message-passing GNNs.

This is a valuable effort that, unfortunately, seems to need further theoretical grounding and results. Reviewers report issues with the theoretical definition of graph reconstructability. The authors clarified in the rebuttal that existing notions are too brittle, that they do not consider partial reconstruction and the role of features. While true, reviewers were concerned that the paper's definitions on the ability of methods to encode node labels and edges without considering how they are jointly encoded, i.e., whether a learned embedding is able to predict topological information/properties (such as cycles, cliques, paths, etc.). Moreover, reviewers argued that little attention is paid to permutation-invariant representations and graph-level tasks. I feel graph-level encoding and edge prediction are not that disconnected, since bi-connectivity is a global graph property tied to predicting the existence of a special type of edge on a graph. But I agree that being able to predict edge marginals (independently) is not a strong case for the reconstruction power of a method. A predictor may be quite good a predicting edges in average and simultaneously be abysmal at predicting higher-order structures.

Overall, I agree with reviewers that the paper needs a more solid theoretical foundation and would advise the authors to follow reviewer recommendations.

**Justification For Why Not Higher Score:**

The paper did not reach the bar necessary for a methodological contribution in GNN expresiveness.

**Justification For Why Not Lower Score:**

N/A

---

### Decision · Program_Chairs · 2024-01-16

Reject